# Single-cell profiling of penta- and tetradactyl mouse limb buds identifies mesenchymal progenitors controlling digit numbers and identities

Victorio Palacio [1,3], Anna Pancho [1,3], Angela Morabito [1], Jonas Malkmus [1], Zhisong He [2], Geoffrey Soussi[1], Rolf Zeller [1], Barbara Treutlein [2] & Aimée Zuniga [1] ✉

The cellular interactions controlling digit numbers and identities have remained largely elusive. Here, we leverage the anterior digit and identity loss in *Grem1* tetradactyl mouse limb buds to identify early specified limb bud mesenchymal progenitor (LMP) populations whose size and distribution is governed by spatial modulation of BMP activity and SHH signaling. Distal-autopodial LMPs (dLMP) express signature genes required for autopod and digit development, and alterations affecting the dLMP population size pre-figure the changes in digit numbers that characterize specific congenital malformations. A second, peripheral LMP (pLMP) population is anteriorly biased and reduction/loss of its asymmetric distribution underlies the loss of middle digit asymmetry and identities in *Grem1* tetradactyl and pig limb buds. pLMPs depend on BMP activity, while dLMPs require GREM1-mediated BMP antagonism. Taken together, the spatial alterations in GREM1 antagonism in mouse mutant and evolutionarily diversified pig limb buds tunes BMP activity, which impacts dLMP and pLMP populations in an opposing manner.

Elucidating the molecular events underlying vertebrate limb buds is shedding light both on our understanding of human limb congenital malformation and evolutionary limb diversification. A key event in limb buds is the activation of Sonic Hedgehog (SHH) signaling by the posterior mesenchymal organizer that is required for both ante-roposterior (AP) patterning and subsequent survival and proliferative expansion of mesenchymal cells[1–3]. Genetic studies show that *Shh* is required in early mouse limb buds (E9.75–E10.0) to specify posterior digits by direct short-range signaling[1,3]. Previous studies indicated that posterior digit specification also depends on the time mesenchymal cells are exposed to *Shh*, while the specification of the anterior digit 2 depends on long-range SHH signaling[4–6]. Yet, the AP sequence and identities of digits are determined only during hand plate (autopod)

formation by the BMP/SOX9/WNT Turing system and signals acting on the distal-most region of the developing digit primordia[7–10]. This reveals a significant knowledge gap in understanding how early digit specification[3] is linked to the determination of the definitive digit pattern and identities. Notably, single cell and lineage analysis of mouse limb buds has identified an *Msx1*[+] cell population of naïve mesenchymal progenitors that progressively transition to proximal and autopodial progenitors which subsequently differentiate into *Sox9*[+] osteochondrogenic progenitors (OCPs)[11]. In addition, it has been shown that *Hoxa13* marks the autopodial lineages that will give rise to metacarpals and digit phalanges[12].

In addition to SHH, BMP signaling is required for digit patterning and development[13–17]. One key event is the activation and upregulation

[1]Developmental Genetics, Department of Biomedicine, University of Basel, Basel, Switzerland. [2]Department of Biosystems Science and Engineering, ETH Zürich, Basel, Switzerland. [3]These authors contributed equally: Victorio Palacio, Anna Pancho. ✉e-mail: aimee.zuniga@unibas.ch

of the BMP antagonist *Gremlin1* (*Grem1*) by BMPs[18] and SHH in the posterior-distal limb bud mesenchyme during the onset of limb bud development[19,20]. This results in the rapid establishment of the self-regulatory SHH/GREM1/AER-FGF feedback signaling system that promotes limb bud outgrowth and mesenchymal cell survival[18,20–24]. An integral part of this self-regulatory signaling system is the spatially dynamic regulation of *Grem1* expression, which preferentially antagonizes SMAD4-mediated BMP signal transduction in the posterior-distal limb bud mesenchyme[25,26] and is essential for digit development[21,27]. The spatial robustness of *Grem1* expression is paramount to normal limb and digit development as genetic reduction of its spatial domain and posterior expression bias causes a transition from wild-type pentadactyly (5 digits) to tetradactyly (4 digits)[28]. In *Grem1* tetradactyl mouse mutant limbs, the anterior digit loss is paralleled by loss of middle digit asymmetry, which shifts the median axis from digit 3 (mesaxonic) to a paraxonic position in-between the two symmetrical middle digits (dotted lines, Fig. 1a). Paraxony is a defining feature of unguligrade posture of the evolutionarily diversified limbs in Artiodactyla[29,30].

In this study, we leverage the anterior digit and asymmetry loss in *Grem1* tetradactyl mouse limb buds to identify two early specified limb bud mesenchymal progenitor (LMP) populations. Both LMP populations are derived from *Msx1*[+] progenitors but are differentially modulated by BMP activity. The spatial distribution and size of a distal-autopodial LMP (dLMP) population depends critically on GREM1 antagonism. In different mouse strains with congenital digit malformations, the spatial alteration of *Grem1* expression in limb buds is linked to a reduction or increase of the dLMP population size that underlies the subsequent loss (oligodactyly) or gain of digits (polydactyly). The second, pLMP population is located in the peripheral (i.e. sub-ectodermal) mesenchyme. In wild-type forelimb buds, the pLMP population is anteriorly biased and this is lost in mutant limb buds due to either anteriorly decreased or posteriorly increased BMP activity. The loss of this anterior bias in the pLMP population is paralleled by the loss of middle digit asymmetry and identities.

This together with analysis of different types of limb congenital malformations and evolutionarily diversified pig limb buds establishes that divergence from the archetypal pentadactyl digit pattern and/or loss of middle digit asymmetry is linked to spatial changes in either or both dLMP and pLMP populations from early limb bud stages onward. The initial alterations affecting the balance of GREM1 antagonism and BMP activity and their impact on dLMPs and pLMPs precede the changes in digit numbers and identities causing congenital limb

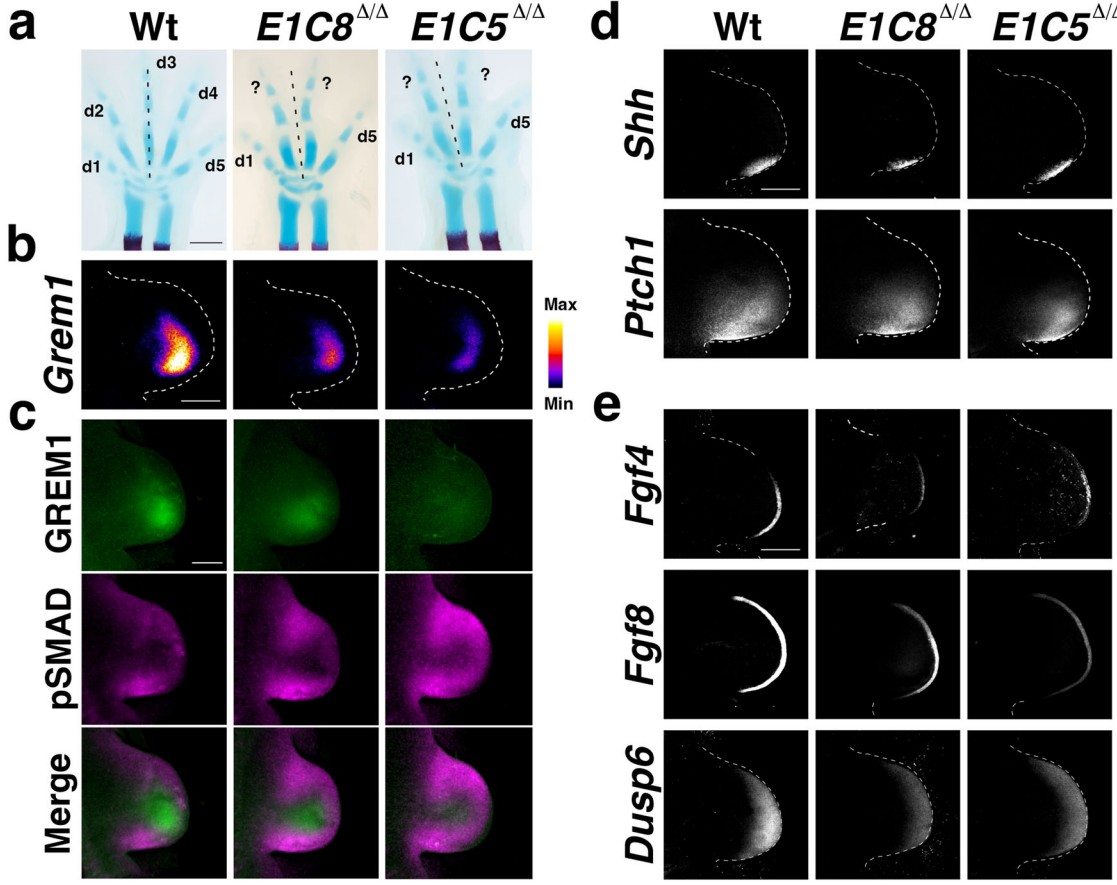

**Fig. 1 | Analysis of the feedback signaling system in wild-type and tetradactyl mouse forelimb buds. a** Skeletal analysis of wild-type (Wt) pentadactyl, *E1C8*[Δ/Δ] and *E1C5*[Δ/Δ] tetradactyl forelimbs at E14.5. Cartilage is stained blue, the ossification centers in the radius and the ulna in red. Digits are indicated from anterior (d1) to posterior (d5). Digits with uncertain identities are labeled by "?". The dotted line indicates the median axis in both pentadactyl and tetradactyl limbs. Per genotype, *n* = 3 independent biological replicates were analyzed. Scale bar: 1 mm. **b** Analysis of the spatial distribution of *Grem1* transcripts by whole-mount RNA-FISH in forelimb buds of all three genotypes at E10.75 (37–39 somites, *n* = 5 replicates per genotype). The transcript levels are shown as intensity values using the fire look-up table of FIJI.

Scale bar: 300 µm. **c** GREM1 and pSMAD activity by whole-mount immunofluorescence in forelimb buds of all three genotypes at E10.75 (*n* = 3 replicates per genotype). Top panels: GREM1, mid panels: pSMAD, lower panels: Merge. Scale bar: 200 µm. **d** Analysis of *Shh* and *Ptch1* expression by whole-mount RNA-FISH in forelimb buds at E10.75 (*n* = 3 replicates per genotype). Scale bar: 300 µm. **e** Analysis of AER-FGF (*Fgf4* and *Fgf8*) expression and mesenchymal FGF signal transduction sensed by *Dusp6* expression in forelimb buds at E10.75 (*n* = 3 replicates per genotype). Scale bar: 300 µm. All forelimb buds are oriented with anterior to the top and posterior to the bottom.

malformations and the appearance of digit reductions in species with evolutionarily diversified limb skeletons.

## Results

### Lineage changes underlying digit 2 loss in *Grem1* tetradactyl mutants

Two different *Grem1* alleles lacking 4 of 7 limb bud CRM enhancers, namely *EC1CRM8*$^{\Delta/\Delta}$ (*E1C8*$^{\Delta/\Delta}$) and *EC1CRM5*$^{\Delta/\Delta}$ (*E1C5*$^{\Delta/\Delta}$)[28] result in tetradactyl limb phenotypes with loss of an anterior digit (Fig. 1a). Analysis of *E1C8*$^{\Delta/\Delta}$ and *E1C5*$^{\Delta/\Delta}$ mouse forelimb buds at embryonic day E10.75 (37–39 somites) using fluorescent whole-mount HCR-RNA in situ hybridization (RNA-FISH) and immunofluorescence[31] establishes that the tetradactyl phenotype is caused by a remarkable spatial reduction of the *Grem1* transcript and protein expression domains (Fig. 1b, c). In particular, the posterior bias and distal-anterior expansion of *Grem1* expression is disrupted in tetradactyl limb buds (Fig. 1b)[28]. This spatial reduction of GREM1 results in the expansion of pSMAD1/5/9 (pSMAD) activities, which points to an increase in BMP signal transduction (Fig. 1c and Supplementary Fig. 1)[32,33].

Furthermore, *Shh* expression and upregulation of *Ptch1* in SHH-responsive cells are reduced (Fig. 1d). Moreover, AER-*Fgf* expression and signal transduction in the distal-most mesenchyme are also reduced as indicated by the FGF target *Dusp6*[34] and lowered pERK levels[35] (Fig. 1e and Supplementary Fig. 1). This shows that the self-regulatory SHH/GREM1/AER-FGF feedback signaling system[18,20,22,23] remains but is scaled down in both *E1C8*$^{\Delta/\Delta}$ and *E1C5*$^{\Delta/\Delta}$ limb buds. These well-defined tetradactyl phenotypes provide a unique opportunity to study the molecular and cellular alterations that underlie tetradactyly and explore their potential relevance to evolutionary digit reductions.

To identify the anterior digit lost in *Grem1* tetradactyl limbs, genetic lineage analysis was performed in wild-type and *E1C5*$^{\Delta/\Delta}$ embryos using specific CRE drivers to permanently activate a fluorescent cell lineage marker in either the posterior or anterior forelimb bud mesenchyme (Fig. 2 and Supplementary Fig. 2). The posterior lineage was traced using the knock-in *Shh*$^{GFPCre}$ allele in combination with conditional activation of the ROSA26$^{LSL-tdTomato}$ reporter in *Shh*-descendant cells (Fig. 2a, b)[6,36]. In wild-type forelimb buds, the *Shh* lineage contributes to the posterior half of the autopod, with most descendants in digits d4 and d5 whereas in digit d3 the lineage is restricted to the posterior part (arrowhead and insets, Fig. 2a)[6]. In *E1C5*$^{\Delta/\Delta}$ forelimb buds, the posterior lineage contribution is comparable to wild type with the highest contributions to digits d5 and d4 and fewer cells to the interdigit between digit d4 and the 3rd digit from posterior (d3*, Fig. 2b; compare insets between Fig. 2a, b).

The anterior lineage was traced using the tamoxifen-inducible *Alx4-Cre*$^{ERT2}$ transgene[37] in combination with the conditional ROSA26$^{LSL-EGFP}$ reporter[38]. The *Alx4-Cre*$^{ERT2}$ transgene is expressed in the anterior mesenchyme up to the limb apex[37]. The anterior lineage was traced by activating the *Alx4-Cre*$^{ERT2}$ transgene at -E10.5–10.75 (Fig. 2c, d). In wild-type forelimb buds, the *Alx4-Cre*$^{ERT2}$ lineage contributes to the anterior half of the autopod (d1, d2, and d3) with descendants being detected only in the anterior part of digit d3 (arrowhead and insets, Fig. 2c; Supplementary Movies S1, S3)[38]. In *E1C5*$^{\Delta/\Delta}$ forelimb buds, the lineage is more restricted due to an anterior shift of the lineage boundary (arrowhead, Fig. 2d). Descendants are detected in the anterior-most digit d1, interdigit 1, and the 3rd digit from posterior (d3*; insets, Fig. 2d, Supplementary Movies S2, S4).

The anterior and posterior lineages of two age- and shape-matched forelimb buds from different embryos were virtually overlapped to visualize the shift in the AP boundary (Fig. 2e). In wild-type forelimb buds, both lineages contribute to digit 3 (upper panels, Fig. 2e). In *Grem1* tetradactyl limb buds, both lineages contribute to the third digit from the posterior, which identifies this digit as d3 and shows that digit d2 is lost (lower panels, Fig. 2e). This is corroborated by the detection of occasional and transient *Sox9*-positive

condensations in the interdigit space between digit d1 and d3 in *Grem1* tetradactyl limb buds (Supplementary Fig. 2). In addition, this analysis shows that the asymmetry of middle digits d3 and d4 is lost in *Grem1* tetradactyl mouse limbs, which contrasts with wild-type pentadactyly.

### Identification of limb bud mesenchymal cell populations with distinct molecular signatures

scRNA-seq analysis of wild-type and *Grem1* tetradactyl forelimb buds at E10.75 (37–39 somites) was performed to identify (1) the mesenchymal progenitor populations in wild-type forelimb buds and (2) the molecular and cellular changes within these populations that are associated to loss of digit 2 and middle digit asymmetry in *Grem1* tetradactyl forelimb buds. Both *E1C8*$^{\Delta/\Delta}$ and *E1C5*$^{\Delta/\Delta}$ forelimb buds were analyzed to focus the downstream analysis on the changes that are consistently and reproducibly detected in both genotypes. The early limb bud stage (E10.75) was chosen as no morphological differences between wild-type and tetradactyl limb buds are apparent prior to autopod development (≤E11.25) and the LMPs contributing to autopod and digit development are likely to arise during this developmental period[11,39]. Comprehensive single-cell datasets for wild-type, *E1C8*$^{\Delta/\Delta}$, and *E1C5*$^{\Delta/\Delta}$ forelimb buds were generated (Supplementary Fig. S3a). After verification, initial processing, and filtering (see Methods), 47432 forelimb bud cells from the 3 genotypes (Wt: 14934 cells, *E1C8*$^{\Delta/\Delta}$: 17958 cells, *E1C5*$^{\Delta/\Delta}$: 14540 cells) were pooled for unsupervised clustering. This identified the limb bud mesenchymal cells as the largest cluster and several smaller clusters containing the other lineages (Supplementary Fig. S3a, Supplementary Data 1). No discernable differences in the contribution of wild-type, *E1C8*$^{\Delta/\Delta}$, and *E1C5*$^{\Delta/\Delta}$ forelimb bud cells to the different lineages were detected (Supplementary Fig. S3b). The vast majority of cells in the limb bud mesenchymal cluster express *Prrx1*, *Pdgfra*, and *Tbx5*, which together with a heatmap of the top-10 enriched genes confirms their limb bud mesenchymal identity (Supplementary Fig. S3c, d)[40–42].

Next, the limb bud mesenchymal cells from all 3 genotypes (Wt: 11155 cells, *E1C8*$^{\Delta/\Delta}$: 13575 cells, *E1C5*$^{\Delta/\Delta}$: 11071cells) were pooled and analyzed by unsupervised clustering, which identifies 11 distinct mesenchymal clusters (Fig. 3a: Wt only, Fig. 4a: all genotypes). Initially, the top-enriched genes expressed in all wild-type mesenchymal clusters (Fig. 3a) were screened for transcriptional regulators and signaling pathway genes with known functions and/or expression patterns during limb bud development[43–45]. These highly enriched genes were used to define distinct molecular signatures for all wild-type clusters (Supplementary Data 2). Four representative signature genes per cluster were then chosen to generate a dot plot expression matrix that provides insight into differential or cluster-specific gene expression (considering the percentage of expressing cells and transcript levels; Fig. 3b). This identified three chondrogenic clusters (C1–C3) that express the key transcription factor (TF) *Sox9* and collagen *Col2a1* in ≥90% of all cells. The largest chondrogenic cluster C1 is enriched in genes that function in proliferating chondrocytes as well as genes expressed in the core limb bud mesenchyme (*Fgf9*, *Sox5/6*, *Tbx15/18*). Clusters C2 and C3 are enriched in TFs that mark proximal chondrogenic progenitors (*Creb5*, *Irx3/5*, *Gria2*, *Ntn1*, *Pbx1*, *5'Hox* genes) and/or are required for scapula development (*Gsc*: cluster 2, *Emx2*: cluster 3; Fig. 3b and Supplementary Data 2, 3 for complete signatures). Of note, most cells in cluster C2 express the *Tgfb2* ligand, which functions in the induction of chondrogenic fates[46].

In addition, four non-chondrogenic mesenchymal clusters (M1–M4) were identified (Fig. 3b and Supplementary Data 2, 3). Two of these mesenchymal clusters, M1 and M2, express proximal genes and are very heterogeneous in nature (gene enrichment signatures are shown in Supplementary Data 2, 3). Cluster M3 is enriched in genes that regulate mesenchymal cell differentiation (*Osr2*, *Fgf18*)[47,48]. In addition, the BMP antagonists *Grem1* and *Smoc1*, and other genes whose expression is restricted to the ventral and/or dorsal

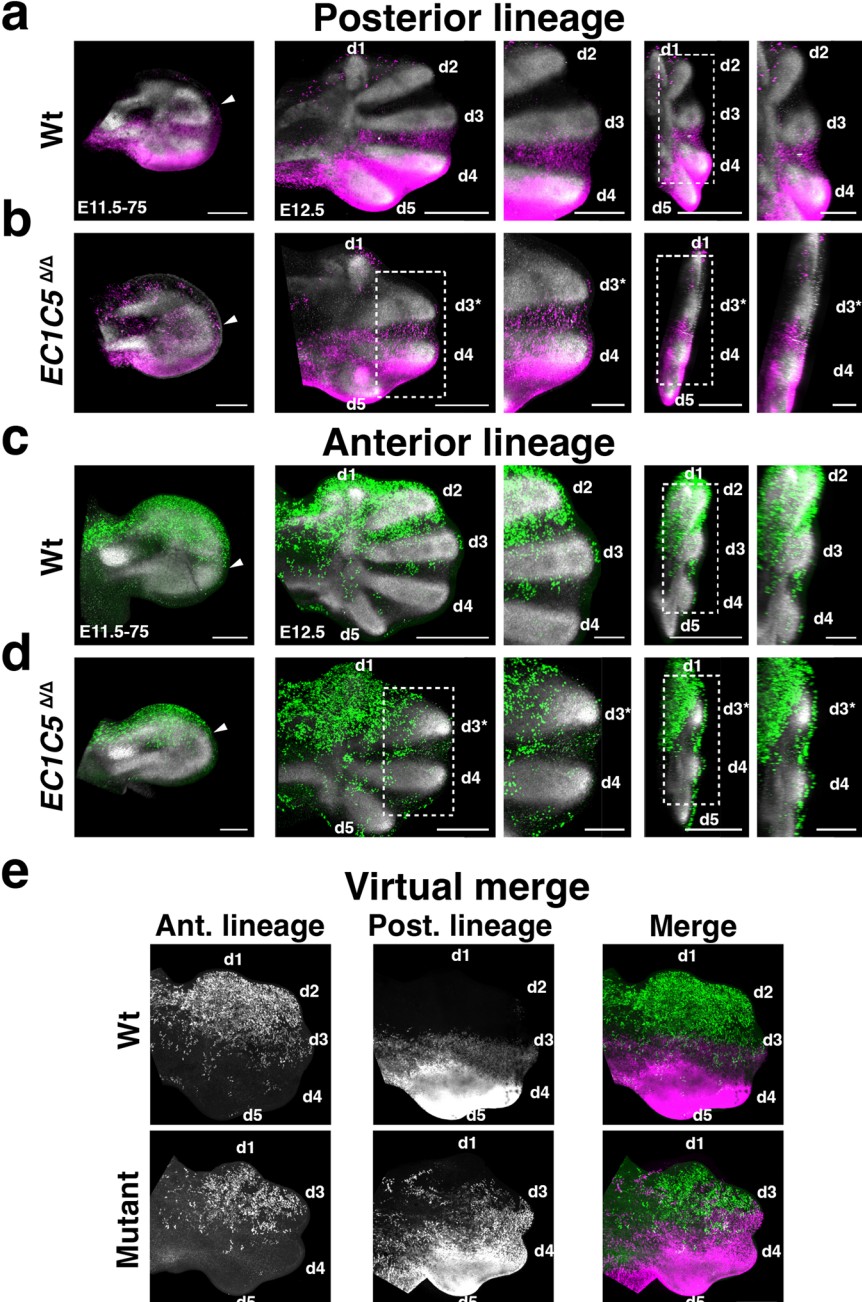

**Fig. 2 | Posterior and anterior lineage analysis of pentadactyl and *Grem1* tetradactyl forelimb buds. a** Tracing the posterior lineage by *Shh*[CRE]-mediated permanent activation of the ROSA26tdTomato reporter (magenta). Representative forelimb buds are shown. Left panel: wild-type forelimb bud at E11.5–E11.75 (*n* = 6/6). Arrowhead indicates the anterior boundary of the posterior lineage. Middle and right panels: wild-type forelimb buds at E12.5 (*n* = 6/6). Enlargements show optical sections to better reveal the posterior lineage between digits d2 and d4. **b** Tracing the posterior lineage in tetradactyl *E1C5*[Δ/Δ] forelimb buds. Left panel: *E1C5*[Δ/Δ] forelimb bud at E11.5–E11.75. Arrowhead indicates the anterior boundary of the posterior lineage, which is shifted in comparison to the wild type (*n* = 3/4). Middle and right panels: *E1C5*[Δ/Δ] forelimbs at E12.5. Enlargements show optical sections to better reveal the posterior lineage between digits d3* and d4. The lineage extends to the middle of d3* (*n* = 5/6). **c** Tracing the anterior lineage by tamoxifen-mediated induction of the *Alx4*-Cre[ERT2] transgene around E10.5, which activates the Rosa26eGFP reporter (green). Representative forelimb buds are shown. Left panel:

wild-type forelimb bud at E11.5–E11.75 (*n* = 6/6). Arrowhead indicates the posterior boundary of the anterior lineage. Middle and right panels: wild-type forelimb at E12.5 (*n* = 5/6). Enlargements show optical sections to better reveal the anterior lineage. **d** Anterior lineage analysis of tetradactyl *E1C5*[Δ/Δ] forelimb buds. Left panel: *E1C5*[Δ/Δ] forelimb bud at E11.5–E11.75 (*n* = 5/5). Arrowhead indicates the posterior boundary of the anterior lineage. Middle and right panels: *E1C5*[Δ/Δ] forelimb at E12.5–75 (*n* = 5/6). Enlargements show optical sections to better reveal the region between digit d3* and d4. Scale bars for (**a**–**d**): 1st panel from the left, 200 µm, 2nd and 4th panels, 500 µm, and 3rd and 5th panels (insets), 250 µm. **e** Virtual merge of the posterior and anterior lineages from two different limb buds at E12.5–75 (*n* = 2). Left and middle panels: single gray channels showing the anterior and posterior lineages after transformation. Right panels: the virtual merge of the anterior and posterior lineages. Scale bar: 300 µm. All limb buds are oriented with anterior to the top and posterior to the bottom, proximal to the left, and distal to the right.

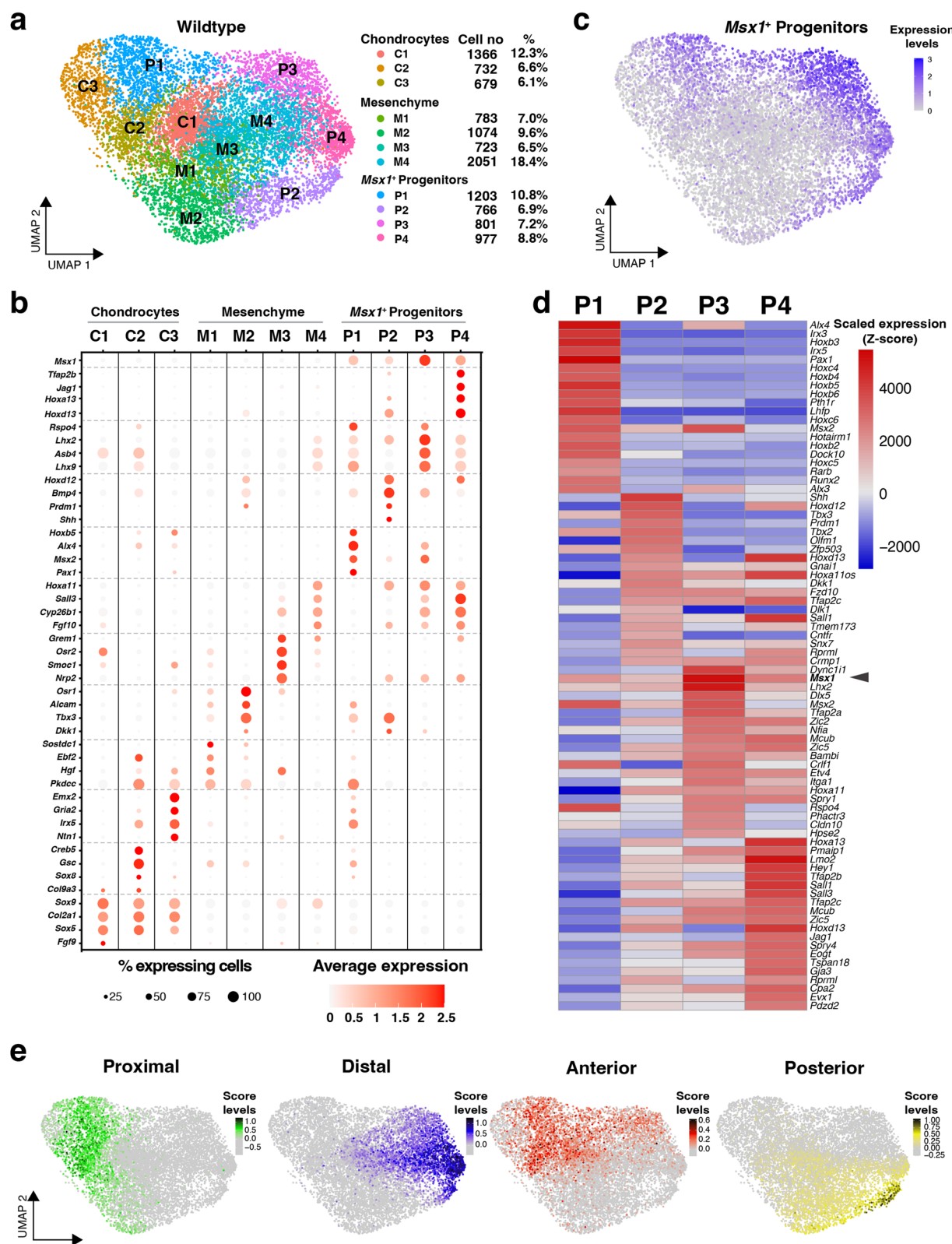

**Fig. 3 | Single-cell RNA sequencing analysis of wild-type forelimb buds (E10.75).** **a** UMAP embedding of the scRNA-seq data from the wild-type genotype. Clusters are annotated to 3 distinct groups (see text): chondrocytes, mesenchyme, and *Msx1*⁺ progenitors. For each cluster, cell numbers and their percentage contribution to all limb bud mesenchymal cells are shown. **b** Dot plot showing the expression of four highly enriched genes with known expression/functions in limb buds for each cluster plus *Msx1* to identify mesenchymal progenitor clusters.

**c** UMAP embedding of *Msx1* shows the *Msx1*⁺ progenitor distribution in wild-type cells. **d** Heatmap showing average expression for the top-20 differentially expressed genes for each of the *Msx1*⁺ progenitor clusters. **e** UMAP embedding of scores using positional markers. Genes used in respective scores were: proximal score: *Rarb, Irx3, Meis2, Pkdcc*; distal score: *Cyp26b1, Sall3, Lmo2, Wnt5a*; anterior score: *Alx4, Boc, Pax1, Sox6*; posterior score: *Hand2, Osr1, Shh, Prdm1*.

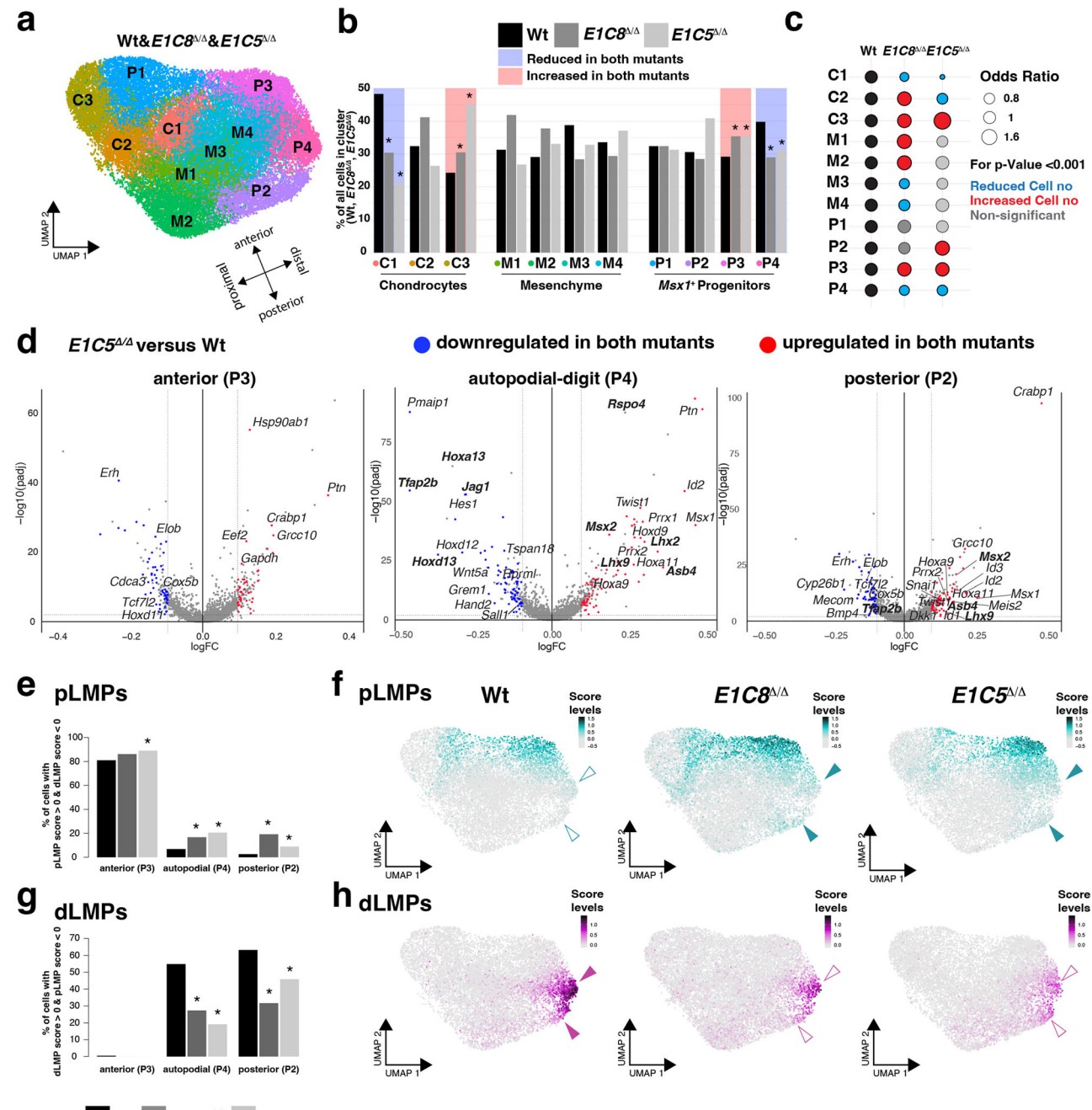

**Fig. 4 | Identification of distinct LMP populations altered in both *Grem1* tetradactyl limb buds (E10.75). a** UMAP embedding of the pooled scRNA-seq data from all three genotypes. Arrows indicate the anteroposterior and dorsoventral axes (see Fig. 3e). **b** Bar plots for all clusters show the percentage of cells for each genotype. Blue underlays indicate clusters with significantly fewer cells in both mutants. Red underlays indicate clusters with significantly more cells Asterisks: *p*-values ≤ 0.001 (two-sided Fisher's exact test). **c** Odds ratios represent differences in mesenchymal cell numbers between wild type and mutants (Wt, *E1C8^{Δ/Δ}*, *E1C5^{Δ/Δ}*). Circle diameter reflects the odds ratio, indicating the probability of a cell being wild type or mutant within a cluster. Red circles signify a higher probability of mutant cells in both *E1C8^{Δ/Δ}* and *E1C5^{Δ/Δ}* clusters (increased cell numbers), while blue circles indicate a lower probability of mutant cells in both mutants (decreased cell numbers). Grey or mixed circles represent non-significant or divergent odds ratios. **d** Volcano plots showing the differentially expressed genes (DEGs) calculated using the Wilcoxon Rank-Sum test in the anterior (P3), autopodial-digit (P4), and

posterior (P2) *Msx1^+* progenitor clusters by comparing wild-type to *E1C5^{Δ/Δ}* limb buds. DEGs expressed in blue are downregulated in both *E1C8^{Δ/Δ}* and *E1C5^{Δ/Δ}* limb buds, while those in red are upregulated in *Grem1* tetradactyl limb buds. Fold changes (x-axis) are shown as log2 values, while significance (y-axis) is log10 of the adjusted *p*-values, i.e., log10(padj). **e**–**h** The analyses shown were generated using the following expression scores for pLMPs: *Asb4*, *Lhx2*, *Msx2*, *Rspo4*; and dLMPs: *Tfap2b*, *Jag1*, *Hoxa13*, *Hoxd13*. **e**, **g** Bar plots show the fraction of pLMP and dLMPs cells as a percentage of all wild-type, *E1C8^{Δ/Δ}* and *E1C5^{Δ/Δ}* cells in the anterior, autopodial-digit and posterior clusters. Cells with a pLMP (**e**) or dLMP (**g**) expression score >0 are shown for all three genotypes. Asterisks: *p*-values ≤ 0.001 (two-sided Fisher's exact test). **f**, **h** UMAP embedding scores for the two LMP populations in wild-type, *E1C8^{Δ/Δ}*, and *E1C5^{Δ/Δ}* limb buds. Coloured arrowheads indicate upregulation and open arrowheads downregulation in mutant forelimb buds. Source data are provided for (**b**, **c**, **e**, and **g**) in the Source Data file.

mesenchyme (*Nrp2, Fmn1, Wnt11, Lmx1b*) are enriched in cluster M3. Cluster M4 is enriched in zeugopodal 3′Hox TFs (*Hoxa/d11*) and distally-expressed genes such as *Sall3, Fgf10, Grem1, Frbz, Cyp26b1* (Fig. 3b and Supplementary Data 2, 3). The enrichment in ligands, antagonists and *Fgfr2* expression shows that mesenchymal cells in clusters M1–M4 are responsive to WNT, BMP, and FGF signaling, suggesting that they are proliferating LMPs[41,42,49].

The remaining four clusters (P1–P4) express the *Msx1* transcription factor, which marks early mesenchymal progenitors (Fig. 3a–d, Supplementary Data 2, 3), that transit progressively to *Sox9*-positive osteochondrogenic progenitors (OCPs) as limb bud outgrowth progresses distally[11]. Cluster P1 is enriched in TFs that function in the anterior-proximal mesenchyme (*Alx4, Hoxb5/3, Msx2, Pax1, Irx3/5*). In addition, *Runx2*, a regulator of osteoblast proliferation and differentiation[50] is enriched highest in cluster P1 (Fig. 3d and Supplementary Data 2, 3). Cluster P2 includes all *Shh*-expressing and a large fraction of SHH-responsive cells (*Ptch1*), the posterior *Bmp2/4* expression domains, and is enriched in TFs that function initially upstream of *Shh* to establish posterior identity (*Prdm1, Hoxd12, Hand2, Tbx2/3*; Fig. 3b, Supplementary Data 2, 3). This establishes P2 as the posterior cluster that encompasses the SHH signaling and the posterior BMP signaling centers. In contrast, cluster P3 is most enriched in *Msx1* (Fig. 3b, d) and TFs required for limb skeletal development (*Lhx2, Lhx9, Msx2*). Additional signature genes include *Asb4* and the secreted WNT pathway activator *Rspo4* (Fig. 3b, Supplementary Data 2, 3), which are expressed with an anterior bias like most P3 signature genes. In addition, the *Bambi* and *Spry1* signaling modulators are also enriched in cluster P3, which points to the regulation of these mesenchymal progenitors by WNT, BMP, and FGF signaling. Finally, cluster P4 expression is highly enriched for TFs (*Hoxa13, Tfap2b, Hoxd13*, Fig. 3b, Supplementary Data 2, 3) that function in autopod and digit patterning[51,52]. The Notch ligand *Jag1* and target TF *Hey1* are also highly enriched in this cluster that encompasses the *Hoxa13*-positive autopodial[12,39] and *Tfap2b*-positive digit progenitors[53] (Fig. 3b, Supplementary Data 2, 3). The transcriptional regulators *Msx1/2, Lhx2/9, Hoxa13/d13,* and *Tfap2b* which are highly enriched in clusters P3 and/or P4 were previously identified as markers for *Msx1*+ progenitors associated with an autopodial genetic program in wild-type limb buds[11].

A heatmap displaying the top-20 enriched genes for clusters P1–P4 (Fig. 3c, d; Supplementary Fig. S4a: heatmap of all 11 clusters) highlights their distinct molecular identities, which is most apparent for the anterior-proximal cluster P1 and the posterior cluster P2. The anterior cluster P3 and autopodial-digit cluster P4 share a fraction of their top-enriched genes; however, the autopodial signature of cluster P4 is very distinct (Fig. 3b, d). Finally, a score for the relevant spatial domains in forelimb buds was generated using specific genes as positional indicators (Supplementary Fig. 4b). Visualization of these scores by UMAP projection shows that unsupervised clustering accurately reproduces the spatial organization of the different limb bud mesenchymal cell types (Fig. 3e, Supplementary Data 3).

### Single-cell transcriptomics identifies two altered LMP populations in *Grem1* tetradactyl forelimb buds

Next, we analyzed the pooled wild-type and two tetradactyl (*E1C8*$^{\Delta/\Delta}$, *E1C5*$^{\Delta/\Delta}$) scRNA-seq datasets (Fig. 4a, Supplementary Fig. 4, Source data) with the aim to pinpoint the cellular and molecular changes that underlie the loss of the anterior digit 2 in both tetradactyl forelimb buds (Fig. 2). Mesenchymal cell numbers were normalized (see "Methods" section) to allow for direct comparison of wild-type and mutant cell clusters (Fig. 4b, Supplementary Data 4). Alterations in cluster cell numbers were considered meaningful when they were either consistently reduced (underlay in blue) or increased (red) in both mutant genotypes versus wild-type forelimb buds (*p*-values < 0.001, Fig. 4b, Supplementary Data 4). The statistical significance of all changes was independently verified by Odds Ratio association testing

(Fig. 4c, see "Methods" section). These analyses identified four clusters with consistent alterations of cell numbers in both *Grem1* tetradactyl mouse forelimb buds at E10.75. Among these, there are two chondrogenic clusters (C1 and C3) with cell numbers reduced in cluster C1 and increased in the proximal cluster C3 (left panels, Fig. 4b, panels C1, C3 in Fig. 4c). These alterations are less pronounced for *E1C8*$^{\Delta/\Delta}$ than *E1C5*$^{\Delta/\Delta}$ chondrogenic clusters (Fig. 4b), which is consistent with more GREM1 protein remaining in *E1C8*$^{\Delta/\Delta}$ than *E1C5*$^{\Delta/\Delta}$ forelimb buds (Fig. 1c). More relevant with respect to digit loss, the other clusters with altered cell numbers are the *Msx1*-expressing clusters P3 and P4. Cell numbers for the anterior cluster P3 are increased, while those for the autopodial-digit cluster P4 are reduced. The cellular alterations in these two clusters could well prefigure the subsequent loss of digit 2 in mutant forelimbs (right panel, Fig. 4b, panels P3 and P4 in Fig. 4c).

To gain insight into the molecular alterations underlying these cellular changes, the differentially expressed genes (DEGs) between wild-type and *Grem1* tetradactyl limb buds were identified for all clusters (P1–P4. Supplementary Data 5; for DEGs in all other clusters: see Source data). Volcano plot visualization identifies the DEGs in the anterior-proximal (P1, Supplementary Fig. 5a), anterior (P3), autopodial-digit (P4) and posterior (P2) *Msx1*+ cell clusters in *Grem1* tetradactyl forelimb buds (E10.75, *E1C5*$^{\Delta/\Delta}$ vs Wt: Fig. 4d, *E1C8*$^{\Delta/\Delta}$ vs Wt: Supplementary Fig. 5b). The most relevant changes in DEGs with respect to signature genes are detected in the autopodial-digit cluster (P4). As expected from the reduction of autopodial and digit progenitor cells (Fig. 4b), the expression of signature genes that include *Tfap2b, Jag1, Hoxa13,* and *Hoxd13* is markedly reduced together with genes like *Hoxd12, Wnt5a,* and *Hand2* (middle panels in Fig. 4d and Supplementary Fig 5b). These downregulated DEGs are expressed and/or function during autopod and digit development. Conversely, in this cluster, *Msx1* and anterior signature genes (*Lhx2, Asb4, Lhx9, Mxs2, Rspo4*, Fig. 3b) are upregulated in both *E1C8*$^{\Delta/\Delta}$ and *E1C5*$^{\Delta/\Delta}$ forelimb buds (middle panels in Fig. 4d and Supplementary Fig. 5b). This is rather unexpected in light of the overall reduced cell number in this cluster (Fig. 4b). In contrast, no differential expression of these genes is detected in the anterior clusters (P3 and P1, left panels Fig. 4d and Supplementary Fig. 5a, b). Finally, in the posterior cluster (P2) *Tfap2b* is reduced, while *Msx1, Lhx2, Asb4,* and *Msx2* expression is upregulated (right panels in Fig. 4d and Supplementary Fig 5b). In addition, several sensors of BMP signal transduction, namely *Id1, Id2* and *Msx2*[54-56] and the SMAD4-target genes[26] *Lhx2, Asb4*[57], *Prrx1/2*[58] and *Rspo4*[59] are upregulated in the autopodial-digit (P4) and posterior (P2) clusters (middle and right panels in Fig. 4d and Supplementary Fig. 5b). Next, upregulated DEGs that are signature genes for the anterior cluster (P3) in wild-type limb buds, namely *Lhx2, Asb4, Msx2* and *Rspo4* were selected to create a score for the LMPs expressing these genes (Supplementary Fig. 5e). The LMP population identified by this approach was named peripheral LMPs (pLMPs, Fig. 4e) due to their largely peripheral distribution by UMAP projection and RNA-FISH (Fig. 4f and Fig. 5, Supplementary Fig. 6). Bar plots based on this expression score reveals the increase of pLMPs in particular in the autopodial (P4:~2–3 fold) and posterior clusters (P2: ~3–8 fold) in tetradactyl *Grem1* forelimb buds (*p* ≤ 0.001, Fig. 4e and Supplementary Data 6). This distal-posterior expansion in the peripheral mesenchyme is also detected by UMAP projections of the pLMP expression score in *E1C8*$^{\Delta/\Delta}$ and *E1C5*$^{\Delta/\Delta}$ forelimb buds (arrowheads Fig. 4f).

Wild-type autopodial-digit signature genes that are downregulated and distally more restricted in *E1C5*$^{\Delta/\Delta}$ and *E1C8*$^{\Delta/\Delta}$ forelimb buds include *Tfap2b, Jag1, Hoxa13,* and *Hoxd13* (Supplementary Fig. 5f), which defines a second LMP population. This population was named dLMPs as its signature score genes function in digit development and/or are expressed in the distal-posterior limb bud mesenchyme (Fig. 6). Bar plot analysis shows that the dLMP population is enriched in the autopodial-digit (P4) and posterior (P2) clusters in wild-type limb buds and significantly reduced in *E1C8*$^{\Delta/\Delta}$ and *E1C5*$^{\Delta/\Delta}$ limb

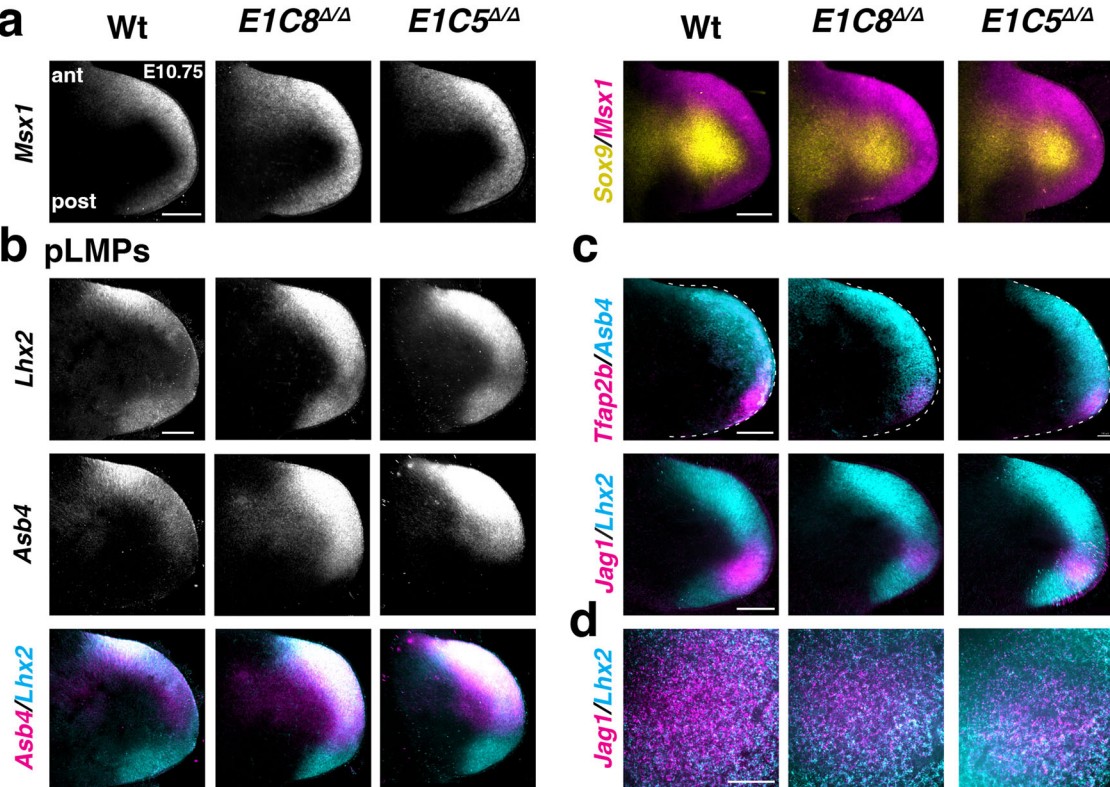

**Fig. 5 | Opposing alterations in pLMPs and dLMPs in *Grem1* tetradactyl limb buds.** Whole-mount RNA-FISH analysis of wild type and *Grem1* tetradactyl forelimb buds. All forelimb buds (E10.75) and higher magnifications are oriented with anterior to the top and posterior to the bottom. Scale bars (**a**–**c**): 200 μm. **a** Spatial distribution of *Msx1*⁺ progenitors (*Msx1*) and OCPs (*Sox9*) in wild-type and *E1C8*$^{Δ/Δ}$ and *E1C5*$^{Δ/Δ}$ limb buds (*n* = 6 replicates for *Msx1* and *n* = 3 for the merge). **b** Spatial distribution of the pLMP signature genes *Lhx2* and *Asb4* in all three genotypes are shown individually in the same forelimb buds (greyscale, top and middle

panels). The overlap of *Lhx2* (cyan) and *Asb4* (magenta) appears white in the highest co-expressing regions (bottom panels) (*n* = 3). **c** Overlap of *Asb4* with *Tfap2b* (top panel) and *Lhx2* (cyan) with *Jag1* (magenta, bottom panel) expression in all three genotypes (*n* = 3). See Supplementary Fig. 6 for individual expressions in greyscale. **d** Optical sections (7 μm) at the level of highest *Jag1* expression in forelimb buds in mutant limb buds. Scale bar: 50 μm. Note that these limb buds are different from the ones shown in (**c**) (*n* = 3). Their whole-mount RNA-FISH patterns are shown in Supplementary Fig. 6 (top panels).

buds (~1.5–2.5 fold, Fig. 4g, Supplementary Data 6). This significant reduction and more distal restriction is also apparent from UMAP projections of the dLMP population in *E1C8*$^{Δ/Δ}$ and *E1C5*$^{Δ/Δ}$ forelimb buds (arrowheads, Fig. 4h). Importantly, the bar plot analysis shows that neither pLMPs nor dLMPs are restricted to one of the four progenitor clusters nor do they constitute the total population of any cluster (Fig. 4e, g) In summary, this unbiased comparative scRNA-seq analysis identifies dLMPs and pLMPs as cellular targets of GREM1-mediated BMP antagonism. Both LMP populations express *Msx1*, which indicates that they arise from *Msx1*⁺ naïve progenitors[11]. DEG analysis shows the differential response of these two distinct LMP populations to BMP signaling and antagonism. In particular, the correct establishment of the dLMP population depends on GREM1 antagonism in the posterior-distal mesenchyme, while the anterior pLMP distribution bias depends on BMP activity (i.e. low BMP antagonism) in the anterior mesenchyme of wild-type limb buds[26].

**Reduction of dLMPs and concurrent increase of pLMPs in tetradactyl limb buds**

To gain insight into the spatial alterations of LMP populations, key signature markers for each population were analyzed by RNA-FISH in forelimb buds at ~E10.75 (~37–40 somites; Figs. 5, 6 and Supplementary Figs. 6, 7). While *Msx1* expression is anteriorly biased in wild-type limb buds, its spatial domain broadens in the distal and posterior mesenchyme in *E1C8*$^{Δ/Δ}$ and *E1C5*$^{Δ/Δ}$ limb buds (left panels, Fig. 5a) in correlation with the increase in *Msx1*⁺ progenitor cells (Supplementary Fig. 5c). Nevertheless, the *Msx1* and *Sox9* expression domains, the

latter marking OCPs and cartilage primordia, remain complementary (right panels, Fig. 5a and Supplementary Fig. 6a). The spatial distribution of the pLMP population was assessed by RNA-FISH analysis of its *Lhx2*[60] and *Asb4*[61] signature genes (Fig. 5b). In contrast to the anterior bias in wild-type limb buds, *Lhx2* expression is increased in the distal and posterior peripheral mesenchyme of *E1C8*$^{Δ/Δ}$ and *E1C5*$^{Δ/Δ}$ forelimb buds (top panels, Fig. 5b). Likewise, *Asb4* expression is highest in the anterior mesenchyme in wild types, while its expression is increased in the distal mesenchyme in both *E1C8*$^{Δ/Δ}$ and *E1C5*$^{Δ/Δ}$ forelimb buds (middle panels, Fig. 5b). The combination of *Lhx2* (cyan) and *Asb4* (magenta) RNA-FISH reveals their co-expression in the same forelimb buds (highest levels appear white, bottom panels, Fig. 5b). The prominent anterior bias in *Lhx2* and *Asb4* distributions (left panels, Fig. 5b) is lost in mutant limb buds owing to expansion of *Lhx2* and *Asb4* expression into the distal-(posterior) peripheral mesenchyme (middle and right panels, Fig. 5b). This expansion agrees with the observed increase of the pLMP population in the autopodial and posterior clusters in both *Grem1* tetradactyl mutants (Fig. 4e).

Next, the spatial expression of select pLMP and dLMP signature genes was comparatively assessed in wild-type and mutant forelimb buds (E10.75, Fig. 5c). Overlapping the expression domains of *Asb4* with *Tfap2b* and *Lhx2* with *Jag1*, respectively, reveals that the distal-posterior expansion of pLMPs is mirrored by reduction of dLMPs in *Grem1* tetradactyl forelimb buds (Fig. 5c, Supplementary Fig 6c). In addition, high-magnification optical sections (7 μm) at the level of the highest *Jag1*⁺ dLMP density of a different set of forelimb buds were analyzed (Fig. 5d and Supplementary Fig. 6d). In wild-type forelimb buds, there is

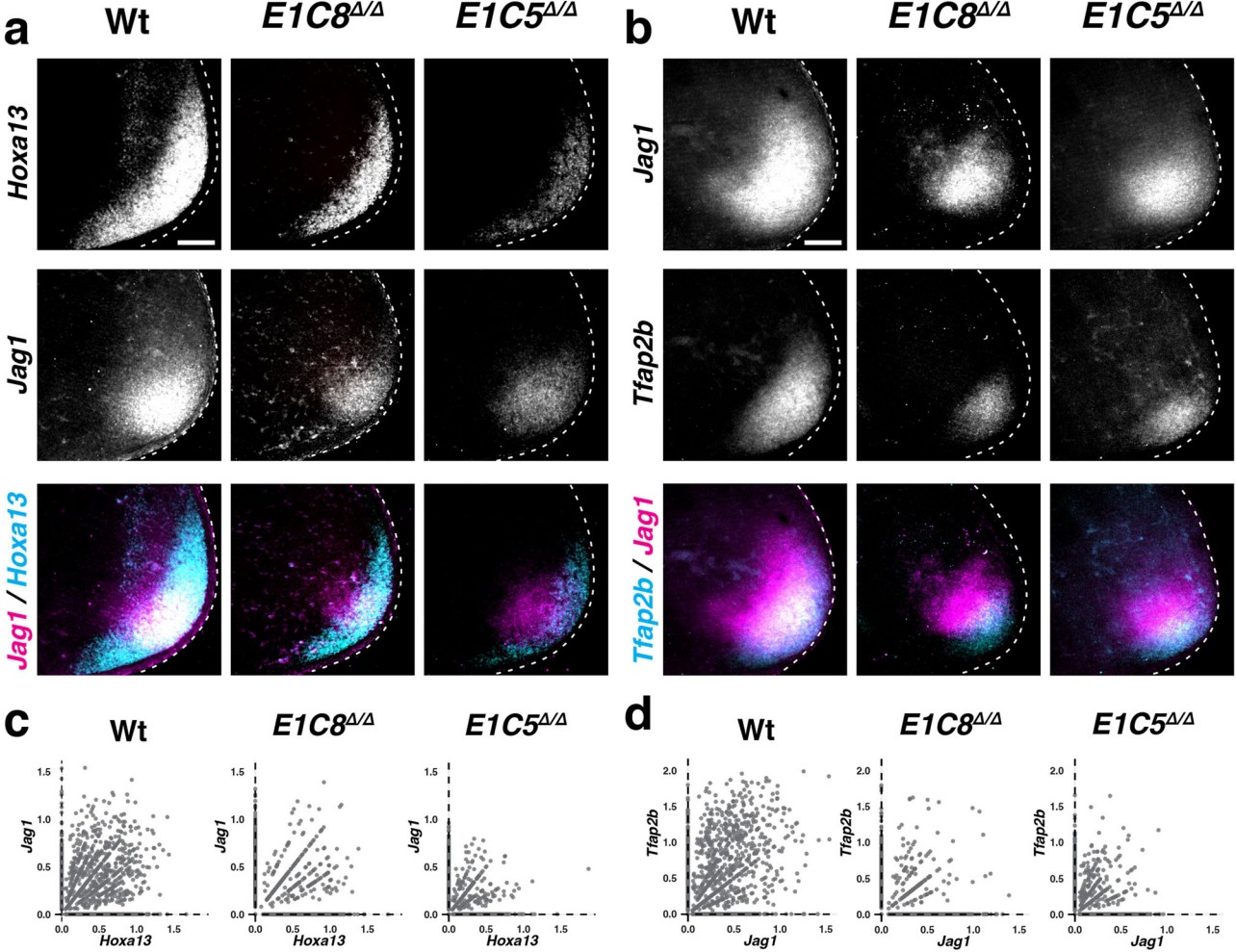

**Fig. 6 | Significant reduction of dLMPs co-expressing *Hoxd13*, *Jag1*, and *Tfap2b* in *Grem1* tetradactyl limb buds. a, b** Whole-mount RNA-FISH analysis of wild-type and *Grem1* tetradactyl forelimb buds. All forelimb buds (E10.75) are oriented with anterior to the top and posterior to the bottom. Scale bars: 100 μm. **a** The spatial distribution of *Hoxa13* and *Jag1* in all three genotypes is shown individually (grey-scale, top and middle panels) and as an overlap (bottom panels) in the same limb buds. This reveals the concurrent reduction of the spatial *Hoxd13* (cyan) and *Jag1* (magenta) expression domains (bottom panels) (*n* = 3 replicates per genotype).

**b** Colocalization of *Jag1* and *Tfap2b* in different limb buds shows that the expression domains of all three dLMP signature genes are spatially reduced in *E1C8^(Δ/Δ)^* and further lowered in *E1C5^(Δ/Δ)^* forelimb buds (*n* = 3 replicates per genotype). **c, d** Scatter plot representation of single mesenchymal cells identifies co-expressing *Hoxa13/Jag1* (**c**) and *Jag1/Tfap2b* (**d**) LMPs in wild type, *E1C8^(Δ/Δ)^* and *E1C5^(Δ/Δ)^* forelimb buds. The number of co-expressing cells is significantly reduced between wild-type and *E1C8^(Δ/Δ)^* and wild-type and *E1C5^(Δ/Δ)^* forelimb buds. Source data is provided in the Source Data file.

a distinct *Jag1*⁺ dLMP domain with some interspersed *Lhx2*⁺ dLMPs in its outer margin (left panel, Fig. 5d). In *Grem1* tetradactyl limb buds, the *Jag1*⁺ dLMP domain markedly reduced while the interspersed *Lhx2*⁺ pLMPs are increased (middle and right panel, Fig. 5d).

The apparent reduction of the *Jag1*⁺ dLMP domain in *Grem1* tetradactyl forelimb buds (Fig. 4g) was analyzed further (Fig. 6). As the *Hoxa13* and *Tfap2b* lineages contribute to digit formation[12,62], their spatial distribution was analyzed by RNA-FISH together with *Jag1* in wild-type and mutant forelimb buds (~E10.75, Fig. 6a, b). The individual RNA-FISH expression patterns (top and middle panels, Fig. 6a, b) are shown together with the colocalization of *Hoxa13* (cyan) and *Jag1* (magenta, bottom panels, Fig. 6a), and *Jag1* with *Tfap2b* (cyan, bottom panels, Fig. 6b). This reveals the extent of spatial co-expression by dLMPs located in the posterior-distal mesenchyme of wild-type and tetradactyl forelimb buds (bottom panels, Fig. 6a, b). The spatial domains of *Hoxa13*, *Jag1*, and *Tfap2b* expression are substantially reduced in *E1C8^(Δ/Δ)^* and *E1C5^(Δ/Δ)^* forelimb buds in comparison to wild-type controls (Fig. 6a, b, Supplementary Fig. 7a).

The spatial reduction observed in mutant limb buds by RNA-FISH was independently confirmed by scatter plot analysis (Fig. 6c, d). This

shows that the number of single mesenchymal cells co-expressing either *Hoxa13* and *Jag1* (Fig. 6c), *Jag1* and *Tfap2b* (Fig. 6d) or *Hoxa13* and *Tfap2b* (Supplementary Fig. 7b) is significantly reduced in *Grem1* tetradactyl forelimb buds in comparison to wild-type controls (Source data: primary data and statistical verification). Furthermore, analysis of forelimb buds between E10.5 (35–36 somites) and ≤E11.25 (40–44 somites) shows that dLMPs are already detected by E10.5 (Supplementary Fig. 8). In *Grem1* tetradactyl forelimb buds, the dLMP population is spatially reduced from the earliest stages onward, which points to a problem in GREM1-dependent specification of the dLMP population size in *E1C8^(Δ/Δ)^* and *E1C5^(Δ/Δ)^* forelimb buds (Fig. 6). In contrast, the early increase and distal-posterior expansion of the pLMP population in mutant forelimb buds (Fig. 5) is transient as their spatial distribution becomes again similar to wild type at later stages (≥E11.0, Supplementary Fig. 8).

## Opposing spatial alterations in dLMPs and pLMPs underlie digit malformations
To gain insight into the extent to which spatial alterations in *Grem1* affect the spatial distribution of dLMP and pLMP signature gene

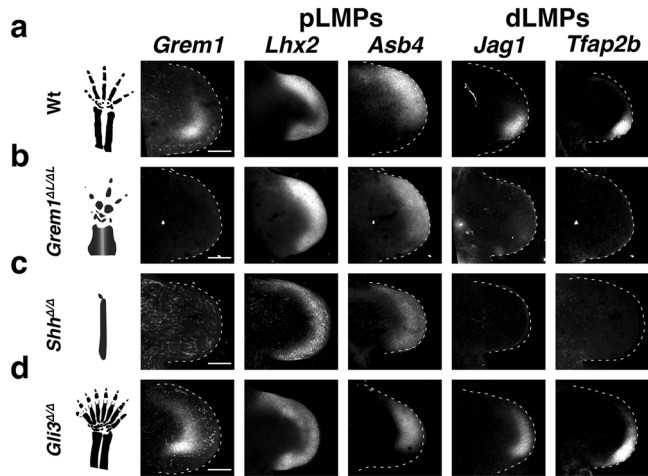

**Fig. 7 | RNA-FISH analysis of *Grem1* and pLMP and dLMP marker genes in mouse models for human congenital malformations.** Comparative analysis of the spatial distribution of *Grem1*; *Lhx2* and *Asb4* (pLMPs); *Jag1* and *Tfap2b* (dLMPs) in wild-type and mutant limb buds. Left-most panel: schemes of the resulting digit phenotypes at E14.5. All limb buds (E10.75) are oriented with anterior to the top and posterior to the bottom. **a** wild-type forelimb buds (*n* = 5). **b** *Grem1*-deficient limb buds (*n* = 2). **c** *Shh*-deficient forelimb buds (prior to the onset of major apoptosis) (*n* = 2). **d** *Gli3*-deficient limb buds (*n* = 4). Scale bars (**a**–**d**): 200 μm.

expression, forelimb buds of mouse models for congenital limb and digit malformations were analyzed (Fig. 7). In mouse forelimb buds lacking *Grem1* (*Grem1^{ΔL/ΔL}*) at E10.75, the pLMP marker *Lhx2* is expressed uniformly within the peripheral mesenchyme, while expression of the *Asb4* marker gene is expanded posteriorly in comparison to wild-type controls (Fig. 7a, b). Strikingly, *Jag1* and *Tfap2b* expression are below detection (Fig. 7b), while the *Hoxa13* expression domain is spatially reduced (Supplementary Fig. 9a). This analysis reveals the predominant dependence of *Jag1^+* and *Tfap2b^+* dLMPs on GREM1 antagonism. Genetic inactivation of *Shh* disrupts *Grem1* expression and results in the formation of one rudimentary digit (Fig. 7c)[20,63,64]. The *Shh* deficiency alters both LMP populations, as *Lhx2* and *Asb4* expression is reduced but uniform within the peripheral mesenchyme, while no *Jag1* and *Tfap2b* expression is detected (E10.75, Fig. 7c). We showed previously that *Hoxa13* expression is reduced to residual levels in *Shh*-deficient limb buds, which agrees with autopod agenesis[65]. Conversely, *Gli3* inactivation results in preaxial polydactyly and precocious anterior expansion of the *Grem1* expression domain (E10.75, Fig. 7d)[65–67]. In *Gli3^{Δ/Δ}* forelimb buds, *Lhx2* expression and the *Asb4* domain are reduced in the anterior part of the peripheral mesenchyme (left panels, Fig. 7d). In contrast, the dLMP markers *Jag1* and *Tfap2b* (right panels, Fig. 7d) follow the precocious anterior *Grem1* expansion together with the anterior expansion of *Hoxa13* expression[65]. This early expansion of dLMPs links to the preaxial polydactyly that arises much later[67], while the anterior reduction of pLMPs is in line with the loss of digit identities in polydactylous *Gli3*-deficient limbs (scheme, Fig. 7d). Collectively, these data indicate that changes in the dLMP population alter digit numbers, whereas loss of the anterior pLMP bias prefigures loss of digit identities.

### Similarities and differences between mouse and pig limb bud development

The symmetrical nature of *Grem1* expression in pig limb buds[28] prompted us to analyze how dLMPs and pLMPs might have been altered during evolutionary diversification of Artiodactyla from the pentadactyl archetype. Therefore, orthologous stages of pig (*Sus domesticus*) limb buds were analyzed using pig-specific HCR probes (Fig. 8). The major difference between *Grem1* tetradactyl mouse and

pig limb buds is that pig limb buds are still pentadactyl at gestational day D30 (scheme, Fig. 8a; orthologous to mouse at E13.25)[68], whereas the loss of middle digit asymmetry and paraxonic axis is already apparent at D24 (Supplementary Fig. 9b; orthologous to mouse at E12.0)[44,68,69]. During digit growth the vestigial anterior digit I is lost due to precocious restriction of AER-FGF signaling[44]. In early pig forelimb buds (D21), *Grem1* is expressed in the posterior mesenchyme comparable to wild-type mouse forelimb buds at E10.5, while pSMAD activity is rather diffuse (Supplementary Fig. 9c). At this early stage, *Msx1^+* LMP and pLMP (*Lhx2*) populations are already expanded into the posterior peripheral mesenchyme, while the dLMP expression domains of *Hoxa13*, *Jag1* and *Tfap2b* are nested in the distal-posterior mesenchyme (Supplementary Fig. 9d–f).

In pig forelimb buds at D23, *Grem1* expression is expanded anteriorly, which is matched by a reduction of pSMAD activity (Fig. 8b). This is paralleled by reduced *Msx1* and anterior *Lhx2* expression (Fig. 8c) in comparison to pig limb buds at D21 and wild-type mouse limb buds at E10.75 (Fig. 5a, b and Supplementary Fig. 9d). In contrast, the dLMP signature genes *Hoxa13*, *Jag1*, and *Tfap2b* remain expressed in spatially-nested posterior-distal domains that scale with the enlarged GREM1 domain (Fig. 8d, e, compare to Supplementary Fig. 8e, f). The analysis of pig limb bud development shows that, as in mouse limb buds, the dLMP population size depends on GREM1-mediated reduction of BMP activity. Similarly, the loss of the posterior *Grem1* expression bias in both mouse tetradactyl and pig limb buds is paralleled by the reduction/loss of the anterior expression bias of *Lhx2* marking the pLMP population. Thus, loss of the anterior pLMP bias in early mouse and pig limb buds appears linked to the loss of middle digit asymmetry in both species.

## Discussion

Two *Grem1* tetradactyl mutants lacking the anterior digit 2 were used as models to study the molecular and cellular alterations underlying digit loss. Single-cell analysis of wild-type and *Grem1* tetradactyl mutant forelimb buds at E10.75 identifies two LMP populations, dLMPs and pLMPs as cellular targets of *Grem1*-mediated modulation of BMP activity. These two distinct LMP populations exhibit gene expression profiles that in large parts overlap the molecular signature of *Msx1^+* progenitors characterized by an autopodial genetic program that was previously identified by analysis of wild-type limb buds[11]. Furthermore, analyzing dLMP and pLMP marker gene expression in limb buds of mouse mutants with congenital loss and gain of digit phenotypes shows that (1) GREM1-mediated BMP antagonism regulates dLMPs and pLMPs in an opposing manner and that (2) altering the size of these LMP populations prefigures changes in digit numbers and loss of middle digit asymmetry. The link between GREM1 and these two LMP populations is relevant as previous studies showed that spatial changes in *Grem1* expression are an early molecular indicator of subsequent alterations of the pentadactyl pattern, whether by congenital malformations or evolutionary diversification of the archetypal pentadactyl pattern[29,30,70,71].

Malkmus et al.[28] showed that conserved CRM enhancers in the *Grem1* genomic landscape[72,73] provide the spatiotemporal regulation of *Grem1* expression and pentadactyly with *cis*-regulatory robustness. Inactivation of several CRM enhancers results in significant spatial reduction and loss of the posterior *Grem1* expression bias from early limb bud stages onward, which causes tetradactyly due to digit d2 agenesis and loss of middle digit asymmetry (this study)[28]. The observed transition from pentadactyl mesaxonic to tetradactyl paraxonic autopod development is reminiscent of the alterations that occurred during the evolutionary diversification of Artiodactyl limbs[28,30,44,68,69,74]. In *Grem1* tetradactyl mouse limb buds, the spatial reduction of GREM1 downscales but does not disrupt the SHH/GREM1/AER-FGF feedback signaling system (this study), whose establishment depends on rapid GREM1-mediated reduction of BMP activity during

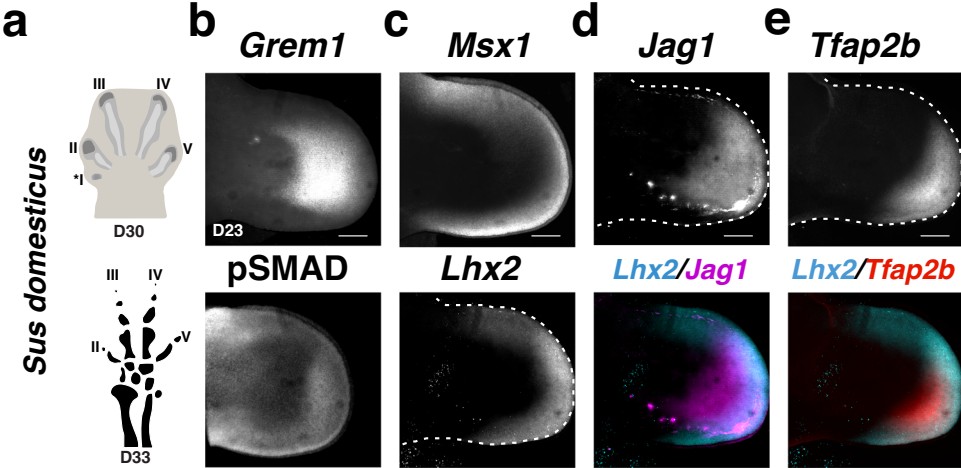

**Fig. 8 | RNA-FISH analysis of *Grem1* and pLMP and dLMP marker genes in pig limb buds.** Analysis of the spatial distribution of *Grem1*, pSMAD, *Msx1*, and signature genes for pLMPs (*Lhx2*) and dLMP (*Jag1*, *Tfap2b*) pig forelimb buds at D23 (*n* = 3 independent biological replicates per probe). **a** Schemes show the pentadactyl digit anlagen in a pig forelimb bud at D30 and the resulting tetradactyl digit pattern at D33 (not drawn to scale, adapted from ref. 44). **b**–**e** Pig-specific HCR probes were used for whole-mount RNA-FISH and pSMAD activity was detected by whole-mount immunofluorescence. Bottom panels in (**d**, **e**) show the merge of the *Lhx2* with the *Jag1* or *Tfap2b* expression domains in the same forelimb bud. All limb buds are oriented with anterior to the top and posterior to the bottom. Scale bars: 300 μm.

onset of limb bud development[18,20,26,75]. The molecular alterations in *Grem1* tetradactyl limb buds and the resulting loss of the anterior digit d2 uncover an additional early function of GREM1 as the postulated relay signal that controls anterior digit specification downstream SHH during early limb bud development[3]. This is further supported by the fact that *Grem1* is required to time the mesenchymal response to SHH signaling in early limb buds[19]. Alternatively, the alterations in *Grem1* tetradactyl limb buds could impact the long-range SHH signaling that has been proposed to pattern digit d2[6].

Indeed, the analysis of *Grem1* tetradactyl and additional mouse mutant limb buds points to an instructive role of GREM1-mediated BMP antagonism in specifying the dLMP population as: 1. the spatial reduction or expansion of the *Grem1* domain directly impacts the corresponding reduction or expansion of the dLMP population in mutant limb buds. 2. maintenance and proliferative expansion of JAG1⁺dLMPs in culture depend critically on inhibition of BMP signal transduction[41]; 3. the expression domains of *Grem1* and dLMP signature genes are proportionally enlarged in pig limb buds compared to mouse limb buds. This shows that across the evolutionary scale, the dLMP population changes in sync with the *Grem1* expression domain. The positive regulation of the dLMP population is likely direct, due to the close proximity of *Grem1*-expressing and dLMP cells. Regulation of the dLMP population size could also involve the SHH/GREM1/AER-FGF feedback loop as genetic inactivation of *Fgf8* together with *Fgf17* or *Fgf9* in the AER results in tetradactyly, but middle digit asymmetry is maintained[34].

That the size of the dLMP population is directly linked to digit number is corroborated by genetic and lineage analysis of signature genes. The *Hoxa13* lineage contributes to all skeletal elements of the autopod including digits[12]. In *Hoxa13*-deficient mouse limb buds, the anterior expansion of both *Grem1* and *Jag1* expression is foreshortened, which is paralleled by tetradactyly and loss of middle digit asymmetry[52,76]. During autopod development, *Jag1* expression is restricted to the distal mesenchyme that will give rise to the digit arch[19,77]. In contrast to *Hoxa13* and *Jag1*, *Tfap2b* expression is more restricted and a recent study shows that the *Tfap2b* lineage contributes to digits[51,62]. Therefore, our single-cell analysis identifies triple-positive *Hoxa13⁺Jag1⁺Tfap2b⁺* dLMPs as early GREM1-dependent digit progenitors in mouse and pig limb buds. Moreover, these dLMP signature genes are expressed by the developing autopod in human embryos[78].

Finally, the changes in dLMP population sizes correlate positively with the numbers of digits that form in mouse embryos with congenital digit loss and gain phenotypes (this study).

In contrast, the pLMP population is increased in the distal-posterior mesenchyme of *Grem1* tetradactyl limb buds from early stages onwards, which shows that the pLMP population correlates positively with increased BMP activity. This is corroborated by the reduction of pLMPs in the anterior mesenchyme of *Gli3*-deficient mouse limb buds as a consequence of the anterior expansion of *Grem1* expression. Indeed, several pLMP signature genes are direct transcriptional targets of SMAD4-mediated BMP signal transduction in the anterior mesenchyme of wild-type limb buds at E10.0[26]. In wild-type forelimb buds, the anterior bias of the pLMP domain is captured by single-cell analysis together with its distal-posterior gap in the region where the dLMP domain is established under the influence of GREM1 antagonism.

Our analysis reveals two molecular alterations underlying the loss of anterior bias in the pLMP distribution. In tetradactyl mouse limb buds, the spatial reduction and lack of posterior bias in *Grem1* expression increases pLMPs in the distal-posterior mesenchyme, which levels out the anterior bias. Conversely in *Gli3*-deficient limb buds, anterior expansion of *Grem1* expression results in loss of pLMP asymmetry due to reduced anterior BMP activity. This in turn promotes the anterior expansion of the dLMPs due to the increased proliferation within the anterior mesenchyme that underlies the preaxial digit polydactyly[67]. In both cases, this causes loss of middle digit asymmetry, which indicates that the anterior bias of the pLMP population is required to specify AP digit polarity in early limb buds. These observations are consistent with the genetic inactivation of the pLMP signature genes *Lhx2/9* in mouse limb buds, which causes loss of AP polarity and digits[60].

Analyzing the evolutionary diversification of the digit pattern in bird embryos has provided evidence that *Grem1* could be an early "sensor" of resulting digit pattern as the size of its expression domain correlates positively with digit numbers in different bird species[79]. Indeed, analysis of the spatial *Grem1* distribution in species from different clades including basal fishes reveals the amazing spatial plasticity in its expression due to the underlying *cis*-regulatory diversification[28]. For Artiodactyla, it has been proposed that spatial alterations in *Grem1* expression prefigure the paraxonic nature and

digit reductions in bovine and pig embryos[28,44,69,74], yet the underlying cellular alterations remained unknown. Pig limb buds are initially pentadactyl but exhibit paraxonic characteristics from the early stages onwards[44,68,69]. Molecular analysis of dLMP and pLMP signature genes in pig limb buds shows that the dLMP population scales with the increase in GREM1-mediated BMP antagonism, unlike the reduction of the *Grem1* domain in *Grem1* tetradactyl mouse limb buds. In contrast, the anterior bias of the pLMP population is lost in both types of tetradactyly: in pig limb buds, the *Lhx2*⁺ pLMP population loses its anterior bias through reduction while in *Grem1* tetradactyl mouse limb buds this bias is lost due to an increase of pLMPs in the distal-posterior mesenchyme. Our analysis supports the conclusion that reduction or loss of the anterior bias in the pLMP population is linked to the loss of middle digit asymmetry, which is a defining feature of unguligrade posture in Artiodactyla[29,30]. Last but not least, the molecular analysis of signature genes concurs with the fossil record which established that the transition from mesaxony to paraxony preceded digit reductions and loss[30,80].

Both the genetic and evolutionary data provide a likely and rather straightforward explanation for the observed significant plasticity of digit numbers and identities in tetrapods. The spatial plasticity in GREM1 spatial regulation[28] can be likened to a control dial that tunes the balance of BMP activity and BMP antagonism. Shifting the balance in either direction would impact one or both LMP populations during evolutionary diversification in ways similar, but not identical to the alterations causing congenital digit malformations and loss. The vast diversity of tetrapod digit patterns even among rather closely related species such as primates[81], Artiodactyla[30], squamates, and birds[79,82] supports the existence of a tunable molecular control system that provides an otherwise robust regulatory system with evolutionary plasticity.

## Methods
### Animals
**Ethics statement and approval of all animal experimentation.** All animal experiments were performed in accordance with national laws and approved by the national/local regulatory and ethic committees/authorities. Mouse studies were approved by the Regional Commission on Animal Experimentation and the Cantonal Veterinary Office of Basel (national license 1950) in accordance with Swiss laws and the 3R principles. France (pig): The study was approved by the local ethical committee for animal experimentation (CEEA VdL, Tours, France). All methods were performed in accordance with the European Communities Council Directive 2010/63/EU for animal protection and welfare used for scientific purposes. Animals were slaughtered in accordance with European regulation under Directive 2010/63/EU in an experimental slaughterhouse with approval number FR37-175-1.

**Mouse strains and embryos.** Mice were housed in individually ventilated cages (Greenline-Tecniplast) at 22 °C, 55% humidity, and a light cycle of 12:12 with 30 min sunrise and sunset. In line with the refine and reduce 3R principles, all strains were bred into a Swiss Albino (*Mus musculus*) background as only robust phenotypes manifest in this strain background and the numbers of embryos and litter sizes are large (≥12–15 embryos per pregnant female). Embryos of both sexes were used for experimental analysis at the developmental stages indicated. Embryos were age-matched by counting somites and matching limb shapes and sizes. The following genetically modified mouse strains were used in this study: two alleles with spatially restricted and reduced *Grem1* expression due to deletion of CRM enhancers in the *Grem1* landscape, namely *EC1CRM5*^(Δ/Δ) (*E1C5*^(Δ/Δ))[28] and *EC1CRM8*^(Δ/Δ) (*E1C8*^(Δ/Δ), this study); for lineage analysis the *Shh*GFPCre (inserted into the *Shh* gene)[6] and *Alx4*-Cre^ERT2 (random transgene insertion)[37] mouse strains were used in combination with the *ROSA26*^[LSL−tdTomato 36] and *ROSA26R* − GFP[38] reporter mice, respectively. *Shh*-deficient embryos were generated

using the *Shh*GFPCre strain and *Gli3*-deficient embryos using the *Gli3*^Δ strain[67]. All genotyping primers are listed in Supplementary Table 1.

**Generation of the *EC1CRM8*^(Δ/Δ) (*E1C8*^(Δ/Δ)) *Grem1* allele.** CRM8 is a highly conserved 736 bp genomic element (present in cartilaginous fish) in open chromatin but LacZ reporter analysis in mouse limb buds did not reveal obvious enhancer activity[28]. Its ancient origin and distal genomic position in proximity to the 3' border and CTCFs sites within the *Grem1* TAD prompted functional analysis in the context of the *EC1* deficiency by genome editing. Two single guide (sg)RNAs were designed to target the 2596 bp region of open chromatin that includes the highly conserved core region of CRM8. The sgRNAs and Cas9 protein were delivered by electroporation to *EC1*^(Δ/Δ)zygotes. Founders were genotyped for the *CRM8* deletion and *EC1*^Δ allele. The accuracy of the *CRM8* deletion was verified by sequencing the breakpoint regions. After breeding founders to the Swiss Albino mice, the deletion of *CRM8* in *cis* to *EC1* was reconfirmed by PCR analysis and sequencing. Initial analysis of *EC1CRM8*^(Δ/Δ) (*E1C8*^(Δ/Δ)) revealed the tetradactyl limb phenotype and spatial reduction of *Grem1* expression (Fig. 1a–c), which resulted in the inclusion of the *E1C8*^(Δ/Δ) allele in this study.

**Pig embryos.** Pig (Sus domesticus) embryos were obtained from artificially inseminated Large White (LW) sows destined for meat production. Embryos were collected at the relevant orthologous stages (D21-D24)[44] at the facility of INRAE Val de Loire (France).

**Skeletal analysis.** For limb skeletal preparations, embryos were collected on embryonic day 14.0, and embryonic stages were assigned using ossification (E14.0–E14.75). Briefly, embryos were collected in PBS and fixed in 95% ethanol (Carl Roth, T171.7) overnight. After staining for 24 h in 0.03% (w/v) Alcian blue (Sigma–Aldrich, A3157), 80% ethanol, and 20% glacial acetic acid (Sigma–Aldrich, 100063) they were washed for 24 h in 95% ethanol. Next, embryos were pre-cleared for 30 min in 1% KOH (Sigma–Aldrich, 105033) and counterstained in 0.005% (w/v) Alizarin (Sigma–Aldrich, A5533) in 1% (w/v) KOH. Finally, embryos were cleared in stepwise increased concentrations of glycerol/1% KOH (20%, 40%, 60%, 80% glycerol) and stored in 80% glycerol in water. Alcian blue and alizarin red detect cartilage and ossified bone respectively. *n* ≥ 3 embryos were analyzed per genotype.

**Fluorescent whole-mount HCR™ RNA in situ hybridization (RNA-FISH).** The mouse HCR™ probes for the different mouse genes analyzed were purchased from Molecular Instruments (USA). Briefly, embryos were fixed in freshly prepared 4% paraformaldehyde (Sigma–Aldrich, P6148) overnight at 4 °C and dehydrated into 100% methanol (VWR, 20903.368) for storage at −20 °C. The RNA-FISH analysis and image acquisition were done exactly as described in the step-by-step protocol[31]. For all RNA-FISH analyses, the number of independent biological replicates is included in the Figure legends, and the limb buds shown in Figs. 1, 5, 6, and Supplementary Figs. 1, 6–8 are representative of the spatial changes observed. There is some minor variation in signal intensities likely due to the rather dynamically evolving gene expression patterns at this forelimb bud stage (~E10.75). For some images acquired by confocal imaging (see below) autofluorescence was removed by image processing. This was done using FIJI to display the maximum projection of the DAPI channel, or substacking to exclude tissue below or above the limb bud (i.e. remnants from dissection and embedding) by generating a selection outside the limb bud. Briefly, this projection was generated using the mean threshold method. Holes were filled in the binary image and if necessary, an iterative binary close or open function was applied to clean the mask. A region outside the limb bud was selected and applied to the original multi-channel stack and pixel values for each channel and slice were set to 0. The selection was saved. Finally, brightness and contrast

were adjusted for each channel. ImageJ macro scripts are available in Zenodo (https://doi.org/10.5281/zenodo.8019804).

**Whole-mount immunofluorescence.** All embryos were dissected in ice-cold PBS and fixed subsequently in 4% PFA overnight. After three 5 min washes in PBS embryos were transferred to 0.1%Tween (Sigma–Aldrich, 93773) in PBS (PBS-T) for splitting embryos in half and stored in 0.01% sodium azide (Sigma–Aldrich, S2002) in PBS at 4 °C. Samples were progressively dehydrated to 100% Methanol in 0.01% PBS-T and subsequent steps were performed as described in ref. [31]. with the following modifications. Incubations with primary antibodies were performed in primary antibody solution: 1% BSA (Sigma–Aldrich, A2153), 10% donkey serum (Sigma–Aldrich, S30), and 0.5% Triton (Sigma–Aldrich, T8787) in PBS-T for 48–72 h with gentle shaking at 4 °C. For anterior lineage analysis, primary antibodies rabbit anti-SOX9 (dilution 1:400, Millipore, Cat# AB5535) and sheep anti-GFP (dilution 1:400, Bio-Rad Cat# 47451051) were used. For posterior lineage analysis rabbit anti-dsRed (dilution 1:200, Takara, Cat# 632496) and goat anti-SOX9 (dilution 1:400, R&D systems, Cat# AF3075) primary antibodies were applied. For GREM1 and pSMAD proteins: primary antibody incubations were performed in primary antibody solution but with 0.4% Triton in PBS. For the detection of GREM1, we used an anti-Human/Mouse Gremlin antibody (dilution 1:100, R&D Systems, Cat# AF956) and for the detection of p-SMAD1.5.9, we used an anti-Phospho-SMAD1/SMAD5/SMAD9 antibody (dilution 1:100, Cell Signaling Technology, Cat# 13820S). For the detection of phospho-ERK1/2 (pERK) and SOX9, the dehydration and bleaching steps were omitted. Primary antibodies used to detect the pERK proteins were rabbit anti-pERK (dilution 1:200, Cell Signalling Technology, Cat# 9101) and goat anti-SOX9 (dilution 1:400, R&D Systems, Cat# AF3075) in primary antibody solution. Following primary antibody incubations, all samples were washed for 6 × 30 min with 0.5% Triton in PBS at room temperature and incubated with secondary antibodies in secondary antibody solution (1% BSA, 10% donkey serum, and 0.5% Triton in PBS) for 48–72 h with gentle shaking at 4 °C. For anterior lineage analysis donkey anti-rabbit 555 (dilution 1:250, Invitrogen, Cat# A31572) or donkey anti-rabbit 647 (dilution 1:250, Invitrogen, Cat# A31573) and donkey anti-sheep-488 (dilution 1:250, Jackson, Cat# 713-545-147) antibodies were used. For posterior lineage analysis donkey anti-rabbit-488 (dilution 1:250, Jackson, Cat# 711-545-152), donkey anti-goat-555 (dilution 1:250, Invitrogen Cat# A-21432) or donkey anti-goat 647 (dilution 1:250, Invitrogen, Cat# A-21447) were used. For GREM1 protein and pSMAD1.5.9. protein the same secondary antibody solution was used with 0.4% Triton in PBS and an incubation of 24 h with the following secondary antibodies donkey anti-goat 647 (dilution 1:1000, Invitrogen, Cat# A-21447) and donkey anti-rabbit 555 (dilution 1:1000, Invitrogen, Cat# A31572). For pERK and SOX9 immunofluorescence donkey anti-rabbit 647 (dilution 1:250, Invitrogen, Cat# A31573) and donkey anti-goat-555 (dilution 1:250, Invitrogen, Cat# A-21432) were applied and were incubated between 48 and 72 h in secondary antibody solution. For all samples cell nuclei were counterstained with DAPI (Sigma–Aldrich, D9542) diluted at 1:1000 into the secondary antibody solution before starting the sample clearing.

**Hydrophilic and hydrophobic clearing of samples for cell lineage analysis.** Hydrophilic samples were cleared in 2.5 M Fructose-Glycerol 60% (v/v) (Fructose: Sigma–Aldrich, F0127; Glycerol: AppliChem, 131339) and mounted as described[31]. Alternatively, hydrophobic tissue clearing was performed as described here (http://www.idisco.info/). Briefly, forelimb buds with flanks were embedded in 2% Agarose (Promega, V3121) prepared in distilled sterile water. Agarose cubes containing the samples were dehydrated through a graded methanol series starting with 20%, ending with 100%, 30 min–1 h each, and an additional step in 100% methanol for 1 h. Samples were placed into 1/3 Methanol and 2/3 dichloromethane (DCM, Sigma–Aldrich, 270997)

with rotation at 13 rpm at room temperature for overnight incubation. The following day, the mixture was replaced with 100% DCM for 30 min with rotation at 13 rpm at room temperature. Subsequently, the 100% DCM solution was removed and dibenzyl ether (DBE, Sigma–Aldrich, 33630) was added by filling the tube to the top to prevent oxidation. Samples were stored in DBE protected from light until imaging. Imaging was performed after refractive index (RI) equilibration for at least 2 h prior to imaging in ethyl cinnamate (ECI, Sigma, 112372) and then placed again in DBE to preserve fluorescence following image acquisition.

**Fluorescent image acquisition and processing.** *Confocal spinning disk microscope acquisition:* after hydrophilic tissue clearing, limb buds were imaged using a 10× objective (10×/0.45 CFI Plan Apo) with a confocal spinning disc scan unit (Yokogawa Spinning Disk CSU-W1-T2) and a Nikon Ti-E, Hamamtsu Flash 4.0 V2 CMOS camera. The image acquisition software VisiView Premier was used to set the acquisition parameters. Z-step size was set to 5 µm for the lineage analysis and to 2.3 µm for RNA-FISH to generate the 2-dimensional maximum projection. The raw *nd* file was converted to an *ims* file using the IMARIS file converter (9.9.1) and by placing the correct voxel sizes. If necessary, auto-fluorescent blood vessels were removed as described below.

*Light-sheet fluorescence microscope acquisition*: after hydrophobic tissue clearing, Zeiss Lightsheet 7 microscope with the ZEN black 3.1 LS (version 9.3.10.393) software and lasers at a fixed wavelength of 405 nm, 488 nm, 561 nm, and 638 nm was used. Dual side illumination was performed using the illumination air lenses LSFM foc 10×/0.2 and adjusted for an RI of 1.55. Fluorescence was detected using a Clr Plan-Neofluar 20×/1.0 immersion detection objective, with the correction collar adjusted to the refractive index. Agarose-embedded samples were adhered to a metal holder and mounted onto the sample holder. Then, the samples were immersed into a 20× clearing chamber filled with 35–45 mL of ECI. Manual alignment of the light sheet was performed using the 561 nm laser and adjusting the collars to the refractive index. The Z-stacks were acquired with the full pco.edge cameras chip (1920 × 1920 pixels), zoom 0.36, using the optimal step size of 1.27 µm to achieve the recommended Nyquist sampling. All acquired images were aligned and fused with Fiji and custom scripts (https://doi.org/10.5281/zenodo.8019804) using the BigStitcher plugin (https://doi.org/10.1038/nmeth0610-418). The results were then converted to an IMARIS file that allows the reconstruction of the 3D volumes using Imaris (Bitplane, 9.9.1). If necessary, auto-fluorescent blood vessels were removed in IMARIS by generating a surface using the 647 channel for autofluorescence (icx) and by masking the generated blood vessel surface in the 488 channel.

**Image processing for cell lineage analysis.** The section view in Imaris was used to visualize the data along the three coordinate axes using the mean intensity projection. Section thickness and locations were defined by setting the xy view from digit d1 to digit d3 in the wild type, from digit d1 to digit d4 in *Grem1* tetradactyl limb buds (anterior lineage), from digit d5 to digit d3 in the wild type, and digit d5 to digit d3* in mutant limb buds (posterior lineage). Section thickness was set from the yz view and covered the whole thickness of the SOX9 digit domain. RGB TIF files were generated with Imaris and the two section views xy and yz were used. OMERO [Software] (https://www.openmicroscopy.org/omero/) was used to generate the insets of these two views. Virtual merge of anterior and posterior lineages in limb buds for better visualization of the domains. For this, age- and shape-matched forelimb buds of anterior and posterior lineage samples acquired by confocal microscopy were used. We employed Fiji (version App 2023-02-172.14.0/1.4f)[83] to create anterior and posterior maximum intensity projections of matched forelimb buds using the DAPI channel and scaled and orientated them to match well. The larger image was scaled down to fit the smaller one (in number of image

pixels). The posterior DAPI channel was aligned with the anterior one using the Fiji plugin MultiStackRegistration[84] in combination with scaled rotation transformation. The generated transformation file was then used to align the two channels of interest for the anterior and posterior lineages. Finally, the transformed anterior and posterior channels were merged, and brightness and contrast were adjusted. An ImageJ macro script is available in the Zenodo repository.

**Processing of limb buds for scRNA-seq.** Mice were set for timed matings and embryos were collected in ice-cold PBS at E10.75 (37–39 somites). Pairs of forelimb buds were dissected from selected embryos and placed into Eppendorf tubes ($n = 3$ biologically independent replicates per genotype). For enzymatic digestion, PBS was removed and exchanged for Collagenase D (Sigma–Aldrich, 11088858001; 20 mg/ml) in DMEM (Thermo Fisher, 41966-029) and incubated in a water bath at 37 °C for 20 min. Limb buds were pipetted gently using a P1000 every 5 min to help dissociation. Enzymatic digestion was stopped by adding 500 μl of HBSS+ (HBSS, 5% Fetal Bovine Serum, 1% HEPES (Thermo Fisher, 15630-080), 1% Pen-Strep (Thermo Fisher, 15140-122)) and the cell suspension was filtered and transferred to a new tube using a FlowMi Cell Strainer (Sigma–Aldrich, BAH13600040) (40 μm) in combination with a P1000 tip. Tubes were centrifuged for 5 min at $2000 \times g$ (4 °C). Cells were resuspended in 100 μl HBSS+ and cell viability and numbers were assessed using a Trypan Blue (Thermo Fisher,15250061) assay on the cell counter. Libraries were generated immediately by the University of Basel Life science genomics facility. For 3 independent biological replicates of wild-type, $E1C8^{\Delta/\Delta}$, and $E1C5^{\Delta/\Delta}$ forelimb buds, libraries were prepared using the single-cell library preparation system Chromium X (10X Genomics) and using the Next GEM Single Cell 3′ Reagent Kit v3.1 according to manufacturer instructions. Verified libraries were pooled and sequenced using NovaSeq 6000 from Illumina.

**Analysis of the primary sequencing data.** We used CellRanger (v.6.0.1) with the default parameters to obtain transcript count matrices aligning the sequenced reads to the mouse genome and annotations. Count matrices were further processed using the R package Seurat (v.5.1.0). In the first instance, cells were filtered on the basis of the number of features expressed, number of counts, and percentage of mitochondrial genes expressed. The threshold for features was greater than 2500, for reads was less than 200,000, and for mitochondrial fraction was below 0.1. All samples were merged into 1 object (merge()) and counts were normalized (NormalizeData). To find and remove doublets from the primary datasets, the DoubletFinder (v.2.0.4) package was used. Subsequently, counts were z-scaled regressing out cell cycle and orig.ident (ScaleData). Principal Component Analysis (Seurat function RunPCA) was performed to obtain a two-dimensional representation of the data, RunUMAP was performed for dimensions 1:30, and otherwise default parameters were used. To cluster cells, the functions FindNeighbors and FindClusters were used, the latter with a resolution of 0.1 to display the major cell types in forelimb buds. FindAllMarkers was applied to the object to identify specific markers for all clusters. Based on these markers, data were subset to restrict the subsequent clustering to the cells that express mesenchymal markers (Supplementary Fig. 3) and filter out cells with mixed lineage marker expression. The remaining mesenchymal cells were clustered using the functions FindNeighbors and FindClusters, the latter with a resolution of 0.8 that yielded 11 distinct clusters (Fig. 3). Module scores for positional and progenitor populations were determined using the function AddModuleScore and the UMAPs color scales show only positive values in a range from minimal to highest positive scores. UMAPs for the expression of individual genes were done using the option order = TRUE unless stated otherwise. To identify differentially expressed genes (DEGs)

between wild-type and mutant mesenchymal cell clusters (wild type versus $E1C8^{\Delta/\Delta}$ and wild type versus $E1C5^{\Delta/\Delta}$) in the $Msx1^+$ mesenchymal progenitor clusters, the pairwise comparisons were conducted using the Wilcoxon rank-sum test. The significantly altered and shared DEGs were visualized in Volcano plots (Fig. 4 and Supplementary Fig. S5). For this DEG analysis, genes located on X and Y chromosomes, mitochondrial, ribosomal, and genes functioning in the hematopoietic system were excluded. For bar plot and scatter plot representations, all cells with ≥0 expression level scores were considered and their counts were incorporated into the bar plot and scatter plot displays.

### Statistical analysis
Bar plots and scatter plots: the statistical significance of the observed differences was assessed using Fisher's exact test, and $p$-values ≤ 0.001 are considered significant. Odds ratio analysis: the association between two categorical variables, namely wild-type versus mutant cell numbers in a cluster, is determined by comparing the odds of an outcome occurring in the wild type to the odds of it occurring in the mutant. An odds ratio of 1 point to no significant differences (indicated in grey), while a ratio greater than 1 suggests an increase in mutant cell numbers compared to the wild type (indicated in red, Fig. 4c). An Odds ratio of less than 1 points to reduced cell numbers in the mutant cell population (indicated in blue). Fisher's exact test uses these associations to determine if the observed distributions are statistically significant ($p$-value ≤ 0.001). DEG analysis comparing wild-type and mutant mesenchymal clusters: Wilcoxon's rank-sum statistical test was used. Statistical Testing was done pairwise for wild-type versus $E1C5^{\Delta/\Delta}$ and wild-type versus $E1C8^{\Delta/\Delta}$ limb buds.

### Reporting summary
Further information on research design is available in the Nature Portfolio Reporting Summary linked to this article.

## Data availability
The Single-cell RNA-seq raw and processed datasets generated in this study have been deposited in the GEO database under accession code GSE267005. Source data are provided as a Source Data file. Source data are provided with this paper.

## Code availability
All original codes have been deposited at Zenodo: https://zenodo.org/doi/10.5281/zenodo.11243709.

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

## Acknowledgements

The authors would like to thank A. Offinger and her team for outstanding mouse husbandry. Additionally, we would like to thank P. Pelczar from the Center for Transgenic Models (CTM), University of Basel, and his team for the generation of genome-edited founder mice. We are grateful to N. Tichy for the whole-mount in situ hybridization analysis shown in Supplementary Fig. 2. We are indebted to V. Tissières for providing the original data for creating the pig limb bud schemes. We would like to thank P. Lorentz from the DBM Microscopy Core Facility for technical support throughout the project and L. Sauteur for providing support for image processing. In addition, we are grateful to the Biozentrum Imaging Core Facility for support with light sheet fluorescence microscopy, especially to S. Roig, K. Schleicher, and L. Gerard. Sequencing was performed at the Quantitative Genomics Facility of the University of Basel and ETH. Calculations were performed using the Scientific Computing Center sciCORE (http://scicore.unibas.ch/) at the University of Basel. The sciCORE team is thanked for support in the curation and storage of the genome-wide datasets. We thank B. Yoder for providing the *Alx4*-Cre[ERT2] mouse. We are grateful to T. Aguirre-Lavin for providing us with pregnant sows and the lab space to collect embryos. This research was supported by grants from the ERC advanced grant INTEGRAL ERC-2015-AdG; Project ID 695032 (to R.Z.), the Swiss National Science Foundation (SNSF): 310030_166685B to R.Z. and A.Z., 310030_184734 and 310030_207824 to R.Z. with A.Z. as a project partner, 310030_192604 to B.T., the National Center of Competence in Research Molecular Systems Engineering to B.T. and the University of Basel provided core funding (to A.Z. and R.Z.).

## Author contributions

A.Z. and R.Z. conceived and supervised the study. Figures were prepared and the manuscript was written by V.P., A.P., A.M., R.Z., and A.Z. with input from all authors. J.M. generated the *Grem1*$^{\Delta E1C8}$ mouse mutant allele. A.P. performed the lineage analysis. A.M. and A.P. performed the immunostainings. V.P. performed the single-cell experiments, and generated and curated the datasets. He also performed the bioinformatics analysis with advice and input from Z.H. and B.T., and together with A.Z. interpreted the single-cell data. V.P., A.M., and A.P. performed the RNA-FISH analysis with contributions from G.S. A.Z. and R.Z. collected pig embryos and A.M. performed the analysis of pig limb bud development. All authors discussed the results and gave input on the manuscript.

## Competing interests

The authors declare no competing interests.
