## [Peer Review file · Nature Communications]

Single-cell profiling of penta- and tetradactyl mouse limb buds identifies mesenchymal progenitors controlling digit numbers and identities

Corresponding Author: Professor Aimée Zuniga

Version 0:

Reviewer comments:

Reviewer #1

(Remarks to the Author)

A number of signaling cues modify digit number and identity within the developing limb bud, however, how these signals converge to specify these properties is poorly understood. Modulation of WNT, HH, BMP, FGF signals all impact these processes. With more primitive lineage tracing methods, the early limb mesenchymal progenitor (LMP) population has been shown to exhibit degrees of heterogeneity. The manuscript by Palacio et al., uses a combination of genetic lineage tracing in various genetic backgrounds coupled with scRNA-seq analysis to identify distinct LMP populations that influence limb digit number and identity. In this regard, they nicely show that the distal-posterior LMP population is compromised in a *Grem1* hypomorphic background and this contributes to loss of middle digit asymmetry. They extend these findings further, and show their model is applicable to tetradactyl digit formation in the pig. Collectively, the findings coalesce around the concept that differential LMP responsiveness to BMPs shapes, in part, limb digit identity and number.

The paper is well-written and the data are of high quality. I particularly like the use of the two different Cre lines to study the anterior and posterior limb mesenchyme compartments in various genetic backgrounds. My main concern is that there is an over reliance on qualitative read-outs and these should be bolstered with more quantitative measurements.

Comments:

- 1) For analysis of population size, scRNA-seq has several caveats including differential dissociation, capture efficiency and data filtering across samples. Ideally, the differences observed in "population size" (i.e., percent of cells) shown in Figure 4a would be validated using more quantitative methods. The subsequent FISHs are certainly supportive in some cases, however, this is also mostly qualitative. As these findings are central to the model it would be useful to further validate the proposed differential abundance of these populations (i.e., flow cytometry if specific and appropriate markers exist) or genes enriched in these populations (i.e., RT-qPCR or comparable), especially for clusters 8, 4 and 10.
- 2) Presumably the captured and filtered/analyzed cells are predominantly mesenchymal in nature, do they exhibit expression of typical mesenchymal markers (i.e., *Prrx1*)?
- 3) It would be useful to provide a single aggregate UMAP in which the samples are color-coded.
- 4) Many of the FISH panels indicate an n=3, which image is chosen for inclusion in the panel?
- 5) This is a minor point, but it would be nice to organize the panels from high *Grem1*/*GREM1* to low, i.e., WT, E1C8, E1C5.

(Remarks on code availability)

Reviewer #2

(Remarks to the Author)

General Comments:

The manuscript presents significant findings that enhance our understanding of how the BMP signaling pathway regulates

LMP gene expression and their spatial distribution, ultimately leading to specific digit patterns. These insights are crucial for the field of limb development and developmental biology in general. However, several aspects of the manuscript could be improved to enhance its clarity and comprehensiveness.

1. Cluster Annotations:

The cluster annotations should be consistent throughout the manuscript. It would be beneficial to present these annotations, particularly for groups such as pLMP and dLMP, in Figure 3. Instead of using numbers, I suggest directly using the annotations (e.g., pLMP, dLMP) that are referenced in the text.

2. Transcriptional Identity:

The current presentation of data does not sufficiently convey the transcriptional identity of the identified clusters (e.g., autopod progenitors, naïve progenitors, proximal/autopod chondroprogenitors). I encourage the authors to define and characterize each cluster in the WT samples more deeply. This can be achieved by presenting specific/enriched genes, such as signaling molecules, transcription factors, and receptors, before discussing their spatial location.

3. Transcriptional Changes in Mutants:

After characterizing the WT clusters and their spatial location within the limb, it would be beneficial to demonstrate how these clusters change in mutant samples. This should include analyses of (1) cell numbers, (2) gene expression, and (3) spatial location within the limb. Currently, only cluster 8 DEGs are analyzed between WT and mutants. Expanding this analysis to all clusters would greatly enhance the manuscript's value.

Specific Comments:

1. Distal Clusters (Msx1+ progenitors):

The rationale behind the statement that Msx1+ progenitors contribute to skeletal primordia is unclear. Are the authors using this as evidence that Msx1+ cells are progenitors?

2. Supplementary Figure 3b:

The presence of Hbb genes and proliferation genes suggests that the data has not been "cleaned" of irrelevant genes and/or cells. Removing these from the clustering analysis would not significantly alter the results but would provide a more relevant list of genes.

3. Figure 4a:

Please clarify how the percentages were calculated, as the differences are minor. It may be more informative to show the percentage of WT, E1C5 MUT and E1C8 MUT within each cluster to emphasize the differences.

4. Msx1 Expression (Cluster 8):

The current description appears to conflate the expression levels of Msx1 genes in cluster 8 with the number of Msx1-expressing cells. Additionally, the UMAP projection does not add more validity to the volcano plot results, as it lacks statistical testing. The rationale behind transitioning from increased Msx1 expression in cluster 8 to examining the number of Msx1+ cells needs clarification.

5. Peripheral LMP Population:

The terminology regarding the "m+LMP population" is somewhat confusing. A clearer term might be "Msx1+ cells" as several clusters express Msx1, indicating a heterogeneous population. Also, it would be helpful to indicate which clusters are considered peripheral mesenchyme.

6. pLMP Marker Genes:

The authors should explain why Lhx2, Asb4, Msx2, and Rspo4 were chosen as markers for pLMP. For instance, Msx2 and Rspo4 are strong markers for clusters 3 and 10, while Asb4 is expressed in several clusters, raising questions about their specific relevance.

7. Figure 4 and Supplementary Figure 4:

The added value of showing the differences in cell numbers of the pLMP population (Fig.4d) after demonstrating differences in cluster cell numbers (Fig.4a) is unclear.

"pLMP population size is increased in distal and distal-posterior clusters"- the terminology (population size in clusters) is confusing. The data presented shows that in the mutants, the pLMP scores are increased in cells located in the distal and distal-posterior regions within the limb.

"DEGs downregulated in cluster 8 is large restricted to the distal clusters" - Fig. 4a is not relevant to this point. It should probably be Fig. 4b.

Please, provide precise panel references in Supp. Figure 4 .

8. dLMP Signature Genes:

The selection of Tfap2b, Jag1, and Hoxa13 as signature genes for the distal-posterior LMP population raises questions about how these genes were chosen. These genes were previously used as markers for autopodial progenitors (Scotti et al., 2015, Zhang et al., 2023, Markman et al. 2023,). For the sake of clarity and uniformity, the authors might consider using this term.

9. Both LMP populations originate from m+LMPs(38):

Please clarify. To my understanding, dLMPs are equivalent to "Autopodial progenitors" in the cited paper, while pLMPs are likely equivalent to "Naïve progenitors".

10. Msx1 Expression in Mutants:

The authors claim that the Msx1 expression domain in the mutant limb is broader than in the control (Fig. 5a). However, the figure does not clearly represent these differences.

Additionally, Msx1 expression levels between WT and mutants seem inconsistent between the left and the right panels in Fig.5a.

11. pLMP and dLMP Spatial Distribution:

To accurately demonstrate the location of pLMPs using the selected markers, the authors should provide in situ evidence showing the overlap of Lhx2 and Asb4. Similarly, demonstrating the location of dLMPs (Jag1 and Tfap2b) also necessitates in situ evidence of overlap between the chosen markers.

12. Figure 5 (Salt-and-Pepper Pattern):

The current images do not adequately support the "salt and pepper" pattern claim. Increasing resolution and magnification is necessary, along with quantitative validation of dLMP cell numbers and gene expression levels. Providing single-channel images for Figures 5e-f would also be beneficial.

(Remarks on code availability)

Version 1:

Reviewer comments:

Reviewer #1

(Remarks to the Author)

The revised manuscript has been markedly improved and the authors have nicely addressed my earlier concerns. I have noted some minor corrections below that need attention.

Line 160, Supplementary Fig. S3b, should be Supplementary Fig. 3a, c, d (not b).

Line 190, in genes to regulate, replace to with that.

Line 205, there is no Runx2 row in Figure 3b, there is one in Figure 3d.

Line 257, tow, should be two.

Line 291, there is no Fig. 5e.

Line 322, replace in correlates the increase with and correlates with the increase.

Reviewer #2

(Remarks to the Author)

The authors have adequately addressed all my comments, and I fully support the publication of this manuscript in its current form

made.

Response to reviewers

General Comments

We would like to thank both reviewers for the positive evaluation of our study and the constructive input that has allowed us to improve the manuscript very significantly. In revising the manuscript we have fully addressed the comments of both reviewers and conducted the requested additional bioinformatics analysis, quantitation and statistical verification, and more RNA-FISH analysis. This has resulted in major changes to Figures, Tables, and Source data. This is paralleled by extensive rewriting of entire sections of manuscript text to increase clarity and precision (see the enclosed marked manuscript with all main Figures included). Briefly, we have repeated the bioinformatics analysis from the start, incorporating improved filtering methods and providing greater detail on the selection of mesenchymal cells. As a result the clustering is slightly different but this does not alter any of the conclusions. The new **Figure 3 (and Supplementary Fig. 3, 4)** shows the gene enrichment analysis and the fraction of cells expressing key markers for all clusters in **wild-type** forelimb buds. The distinct gene enrichment signatures for all clusters provides insights into their cellular and molecular identities as well as their spatial location in forelimb buds. **Figure 4 and Supplementary Fig. 5** (plus Source Data) includes now a comprehensive analysis of all clusters in wild-type and mutant forelimb buds and statistical analysis of the differences in cell numbers and their DEGs. The second part focuses on the alterations observed in the distal clusters that enriched in *Msx1*-positive progenitors. **DEG analysis** including all four *Msx1*-expressing clusters results on positive identification of two distinct dLMP and pLMP populations. The RNA-FISH analysis was extended in light of the reviewer's comments and now shows that the observed reduction of dLMPs in mutant forelimb buds compares well with the significant quantitative differences observed by scatterplot analysis (revised **Figure 5**, new **Figure 6** and **Supplementary Fig. 6, 7**). Together these revisions reinforce and strengthen the conclusions and increase the understanding of the analysis and coherence of the study.

Reviewer #1

A number of signaling cues modify digit number and identity within the developing limb bud, however, how these signals converge to specify these properties is poorly understood. Modulation of WNT, HH, BMP, FGF signals all impact these processes. With more primitive lineage tracing methods, the early limb mesenchymal progenitor (LMP) population has been shown to exhibit degrees of heterogeneity. The manuscript by Palacio et al., uses a combination of genetic lineage tracing in various genetic backgrounds coupled with scRNA-seq analysis to identify distinct LMP populations that influence limb digit number and identity. In this regard, they nicely show that the distal-posterior LMP population is compromised in a *Grem1* hypomorphic background and this contributes to loss of middle digit asymmetry. They extend these findings further, and show their model is applicable to tetradactyl digit formation in the pig. Collectively, the findings coalesce around the concept that differential LMP responsiveness to BMPs shapes, in part, limb digit identity and number.

The paper is well-written and the data are of high quality. I particularly like the use of the two different Cre lines to study the anterior and posterior limb mesenchyme compartments in various genetic backgrounds. My main concern is that there is an over reliance on qualitative read-outs and these should be bolstered with more quantitative measurements.

We thank this reviewer for insightful comments and agree with all points raised. In response, we have clarified the rationale for analysing two mutant alleles and have incorporated substantial additional quantitative and statistical analyses to further support and substantiate the data presented (see below).

Comments:

For analysis of population size, scRNA-seq has several caveats including differential dissociation, capture efficiency and data filtering across samples. Ideally, the differences observed in "population size" (i.e., percent of cells) shown in Figure 4a would be validated using more quantitative methods.

The subsequent FISHs are certainly supportive in some cases, however, this is also mostly qualitative. As these findings are central to the model it would be useful to further validate the proposed differential abundance of these populations (i.e., flow cytometry if specific and appropriate markers exist) or genes enriched in these populations (i.e., RT-qPCR or comparable), especially for clusters 8, 4 and 10.

We appreciate and have carefully considered this reviewer's suggestion for FACS-based isolation of the specific pLMP and dLMP progenitor populations and analysis of specific genes by RT-qPCR. Unfortunately this is not feasible, as there are no suitable cell markers to isolation of the LMP populations by FACS. We have realised as part of some experiments connected to this study that RT-qPCR using total RNA from limb buds is more variable, i.e. less reliable than RNA-seq. Furthermore dLMPs correspond only to about 12-17% of all cells in wildtype limb buds at E10.75 and in mutants the population size is reduce between 2-3 fold.

However we agree with this reviewer about the potential caveats of scRNA-seq. This is why to address these as much as possible we analysed two independently generated *Grem1* alleles exhibiting similar reductions in *Grem1* expression (**Fig. 1**) that causes loss of digit 2 (**Fig. 2**). In addition, the single cell data for each genotype consist of 3 independent replicates that were pooled only after initial verification to exclude major technical and/or biological variation and the filtering prior to downstream analysis was identical across all samples. Focusing the downstream analysis on the cellular and molecular alterations that are shared by both mutant alleles and statistically significant provides additional data robustness to the analysis as it also minimizes the influence of biological variation. Only statistically verified significant changes are shown and followed up by additional analysis. All gene enrichments (Fig. 3), fractions of expressing cells and DEG analysis focuses on statistically significant differences (**Supplementary Tables 1, 2 and 5**; and **Source data** for p-values). The downstream analysis includes several types of statistically verified quantitative analyses such as bar plots (asterisks indicate significance $p \leq 0.001$) and Odds ratio analysis, and Scatterplots (**Fig. 4, Supplementary Fig. 5, Scatterplots in Fig. 6 and Supplementary Fig. 7, and Source Data**). For the gene enrichment analysis of all clusters and of DEGs not only fold changes in gene expression but also the changes in the percentage of expressing cells were determined, statistically validated and included in the respective Supplementary Tables.

This quantitative approach is apparent throughout the revised manuscript and Figures. For example, for the LMP populations: while the anterior (P3), and autopodial-digit (P4-former cluster 8) and posterior clusters (P4) are enriched in pLMPs and dLMPs respectively, bar plot analysis shows that not all cells within these clusters are pLMPs or dLMPs in wildtype forelimb buds (**Fig. 4e, g**). In both mutants, pLMPs are significantly increased in the autopodial (P4) and posterior (P2) clusters, while dLMPs are significantly decreased in these clusters (asterisks indicate statistical significance).

RNA-FISH analysis is qualitative and some variation is detected (see **Figures for reviewers**). Therefore, we have confirmed the spatial reduction of dLMPs signature genes *Hoxa13*, *Jag1* and *Tfap2b* by quantitative single cell scatterplot analysis, which reveals statistically significant differences in the number of dLMP cells that co-express signature genes in forelimb buds of both *Grem1* tetradactyl mutants (**Fig. 6c, d, and Supplementary Fig. 7b**). For pLMPs, the observed difference in both mutant forelimb buds, namely the posterior expansion of the *Lhx2* and *Asb4* (**Fig. 5**) is matched by the significant increase of the fraction pLMP cells in the autopodial and posterior clusters (**Fig. 4e**).

2) Presumably the captured and filtered/analyzed cells are predominantly mesenchymal in nature, do they exhibit expression of typical mesenchymal markers (i.e., *Prrx1*)?

During the initial filtering process doublet and cells expressing mixed lineage markers were removed. Then, unsupervised clustering of all three genotypes identified mesenchymal cells as the largest cell cluster (**Supplementary Fig. 1, Methods section "Analysis of the primary sequencing data"**). As suggested by this reviewer, we show that the mesenchymal cluster expresses established markers such as *Prrx1*, *Pdgfra* and *Tbx5*, which confirms forelimb bud mesenchymal identity (**Supplementary Fig. 3c**). Therefore, this mesenchymal cluster was used for downstream unsupervised clustering of the different mesenchymal populations (**Fig. 3a, Fig. 4a**)

3) It would be useful to provide a single aggregate UMAP in which the samples are color-coded.

Agreed. These data are now included in **Supplementary Fig. 3b**.

4) Many of the FISH panels indicate an n=3, which image is chosen for inclusion in the panel?

We have repeated and expanded the RNA-FISH analysis (**Figs. 5, 6, Supplementary Figs. 6, 7**), which increased samples sizes (**n=x indicated in all Figure legends**). The images chosen for the Figure panels are representative for the spatial alterations in expression (**see the included Figures for reviewers**). There is some variation, likely due to the dynamically evolving gene expression patterns of pLMP and pLMP signature genes at this early pre-autopod stage that is documented by an RNA-FISH timeline

(**Supplementary Fig. 8**). However, in all cases the spatial expression domains of the dLMP signature genes *Hoxa13*, *Jag1* and *Tfap2b* are reduced while the pLMP signature genes *Lhx2* and *Asb4* are expanded distal-posteriorly in mutant forelimb buds (**Figs. 5, 6 and Supplementary Figs. 6, 7**). These data are also matching the quantitative single cell analysis included in the revised manuscript (**Fig. 4e,g, Fig. 6c, d and Supplementary Fig. 7b**).

5) This is a minor point, but it would nice to organize the panels from high *Grem1*/GREM1 to low, i.e., WT, E1C8, E1C5.

Agreed and done.

Reviewer #2

General Comments:

The manuscript presents significant findings that enhance our understanding of how the BMP signaling pathway regulates LMP gene expression and their spatial distribution, ultimately leading to specific digit patterns. These insights are crucial for the field of limb development and developmental biology in general. However, several aspects of the manuscript could be improved to enhance its clarity and comprehensiveness.

We thank this reviewer for the insightful and constructive comments that we have all addressed in the manuscript by conducting additional analysis. In particular, we now include an in-depth gene enrichment analysis that enables definition of specific signatures for all mesenchymal clusters in wild-type forelimb buds (**Fig. 3 and Supplementary Fig. 4**). As suggested by this reviewer, this provides valuable insight into cellular identities and the responsiveness of specific clusters to the limb bud signaling pathways. It also establishes a solid foundation for downstream comparative analysis of wild-type and *Grem1* tetradactyl limb buds. Furthermore, the bioinformatics analysis shown in **Fig. 4 and Supplementary Fig. 5** has been completely revised to show clearly in which clusters cell numbers are changed and how pLMPs (increased in mutants) and dLMPs (decreased in mutants) were identified by DEG analysis. The analysis shows how pLMPs and dLMPs are distributed among the four *Msx1*⁺ positive clusters in wild-type and how this distribution changes in mutant forelimb buds, which is also seen by RNA-FISH analysis (**Figs. 5, 6 and Supplemental Figs 6, 7**). Below we provide responses to all suggestions /questions.

1. Cluster Annotations:

The cluster annotations should be consistent throughout the manuscript. It would be beneficial to present these annotations, particularly for groups such as pLMP and dLMP, in Figure 3. Instead of using numbers, I suggest directly using the annotations (e.g., pLMP, dLMP) that are referenced in the text.

We agree and have annotated all wild-type clusters using the significantly enriched gene approach (see point 2) suggested by this reviewer (**new Fig. 3 and Supplementary Fig. 4**) and used these for downstream analysis. We also have revised the text and Figures (**new Fig 3, 4 and Supplementary Fig 4, 5**). A defining criteria for the pLMP and dLMP populations are the changes in gene expression (DEGs) between *Grem1* tetradactyl limb buds and wild-type limb buds at E10.75 (**Figure 4d-g and Supplementary Fig. 5**). This is key to our study that aimed to identify the cellular and molecular alterations that underlie the loss of digit 2 in forelimb buds at early stages, i.e. prior to morphological appearance of the autopod primordia. The signature of the pLMP population is defined by DEGs that are upregulated in the autopodal-digit (P4) and posterior (P2) cluster whereas dLMPs are defined by DEGs downregulated in these two clusters. Additionally, both LMP populations are enriched within, but do not encompass all cells in the anterior (P3), autopodal-digit (P4) and posterior (P2) clusters. This distinction is most clearly illustrated in the bar plot analyses (**Fig.4e, g**, violin plots in **Supplementary Fig. 5e, f**). We hope that this is also clearer from the revised description of these results.

2. Transcriptional Identity:

The current presentation of data does not sufficiently convey the transcriptional identity of the identified clusters (e.g., autopod progenitors, naïve progenitors, proximal/autopod chondroprogenitors). I encourage the authors to define and characterize each cluster in the WT samples more deeply. This can be achieved by presenting specific/enriched genes, such as signaling molecules, transcription factors, and receptors, before discussing their spatial location.

Based on this reviewer's suggestions, we have performed the requested gene enrichment analysis and provide gene signatures and the gene enrichment analysis for all 11 mesenchymal clusters in the new **Fig. 3, Supplementary Fig. 4 and Supplementary Tables 2, 3**. These new results are described in the revised **Results** section to allow the readers to understand how the gene enrichment analysis allowed identification of distinct chondrocyte, mesenchymal and *Msx1*-expressing clusters (**Fig. 3 and Supplementary Fig. 4**). As suggested by this reviewer this enrichment analysis of the wild-type dataset identifies the transcriptional identity of all mesenchymal clusters and allows us to connect these findings to previous studies. In particular, the mesenchymal clusters fall into three groups: clusters that are undergoing differentiation to chondrogenic lineages (C1 to C3), mesenchymal cells that are distinct but also rather heterogeneous (M1 to M4) and progenitor clusters (P1 to P4) that express *Msx1*, a marker for early limb progenitor cells (**ref. 11 in manuscript**). These parallels are discussed in detail in the revised **Results** section.

3. Transcriptional Changes in Mutants:

After characterizing the WT clusters and their spatial location within the limb, it would be beneficial to demonstrate how these clusters change in mutant samples. This should include analyses of (1) cell numbers, (2) gene expression, and (3) spatial location within the limb. Currently, only cluster 8 DEGs are analyzed between WT and mutants. Expanding this analysis to all clusters would greatly enhance the manuscript's value.

(1) **Fig. 4b** is now showing changes in cell numbers for all 11 mesenchymal clusters between wild-type and both mutant forelimb buds. Statistically verified significant changes in cluster cell number observed in both mutants (versus wild-type) are indicated in **Fig. 4b (bar plot) and Fig. 4c (Odds ratio analysis)**- please see the description in the text for details.

(2) **Fig. 4d and Supplementary Fig. 5a, b** now show the Volcano plot analysis for all *Msx1*-expressing clusters P1-P4. We agree with this reviewer's suggestion that making DEGs for all other clusters (C1-C3, M1-M4) available provides a valuable resource for the research community. However, to maintain the manuscript's focus and avoid overloading it, the lists of up- and downregulated DEGs for these other clusters have been included in the Source data.

(3) The enrichment signatures of all clusters include genes with known distinct spatial expression in mouse forelimb buds. This allows for straightforward assignment of spatial cue to the clusters. Combined with spatial scores (**Fig. 3e, Supplementary Fig. 4**), these signatures enable mapping of the proximo-distal and antero-posterior axes for UMAP analysis (**Fig. 4a**). These spatially enriched genes are also instrumental in describing the identities of the clusters, DEGs and the pLMPs and dLMP populations. The downstream analysis by RNA-FISH confirms the spatial distributions for genes of interest (**Fig. 5, 6 and Supplementary Figs. 6-8**).

Specific Comments:

1. Distal Clusters (*Msx1*+ progenitors):

The rationale behind the statement that *Msx1*+ progenitors contribute to skeletal primordia is unclear. Are the authors using this as evidence that *Msx1*+ cells are progenitors?

Agreed- We clarify this statement in the **Introduction** section (red marked text) as it is based on the study by Markman et al. (2023, ref. 11). These authors show that *Msx1*+ naïve progenitors transition into proximal and autopodial progenitors that then differentiate into OCPs. We also describe better in **Results** and **Discussion**, how our analysis of wild-type and *Grem1* tetradactyl forelimb buds relates to the analysis of *Msx1*+ progenitors in wild-type limb buds (ref. 11).

2. Supplementary Figure 3b:

The presence of *Hbb* genes and proliferation genes suggests that the data has not been "cleaned" of irrelevant genes and/or cells. Removing these from the clustering analysis would not significantly alter the results but would provide a more relevant list of genes.

We thank the reviewer for pointing this out and have increased the filtering of the primary datasets (**see Methods section "Analysis of the primary sequencing data"**) to remove doublets, cell cycle and cells positive for haemoglobin (*Hbb*), which are likely contaminants from blood cells in limb buds. The revised heatmap is included as **Supplementary Fig. 4a**.

3. Figure 4a:

Please clarify how the percentages were calculated, as the differences are minor. It may be more

informative to show the percentage of WT, E1C5 MUT and E1C8 MUT within each cluster to emphasize the differences.

This analysis was redone to include all clusters as suggested by this reviewer. The new **Fig. 4b** (bar plots) and **Fig. 4c (Odds ratio analysis)** show the differences between the wild-type and both mutants clusters across all clusters. To generate the values shown in Fig. 4b, c we calculated the percentage of expressing cells for each cluster, signature genes and genotype (**Supplementary Tables 3, 4**). Only statistically significant differences (asterisks, p -values ≤ 0.001) observed between wildtype and both mutants (**Fig. 4b, c**) were considered for further analysis of the distal progenitor clusters. To complement the bar plot analysis shown in **Fig. 4b**, Odds ratios were calculated to quantify the changes in cell numbers across all clusters (**Fig. 4c**), which corroborates and supports the differences in cell numbers observed by bar plot analysis.

4. Msx1 Expression (Cluster 8):

The current description appears to conflate the expression levels of *Msx1* genes in cluster 8 with the number of *Msx1*-expressing cells. Additionally, the UMAP projection does not add more validity to the volcano plot results, as it lacks statistical testing. The rationale behind transitioning from increased *Msx1* expression in cluster 8 to examining the number of *Msx1*⁺ cells needs clarification.

The substantial revisions, which include an in-depth analysis of all clusters in wild-type and mutant forelimb buds, should clarify the distinction between gene expression levels and number of expressing cells. We performed two independent analyses. One focuses on gene enrichment and differential gene expression (DEG) to identify fold-changes in gene expression, that are visualized using violin plots, Volcano plots, heatmaps and UMAPs - the latter displaying the distribution of expressing cells (**Fig. 3b-e, 4d,f,h and Supplementary Figs. 4, 5a,b,c (right panel), d-f**). The other quantifies the number or percentage of cells expressing a specific gene or multiple genes (referred to as "score") in wild-type and mutant clusters. This approach provides information on the number/percentage of cells in specific wildtype and mutant clusters and cell populations, such as *Msx1*⁺ progenitors, pLMPs and pLMPs, regardless of expression levels (scores >0). These data are visualized by bar plots or scatter plots, which show changes in cell numbers and distributions (**Fig. 4b,c,e,g, Fig. 6c,d, Supplementary Fig 5c (left panel), 7b**). In summary, changes in gene expression levels and cell numbers are analyzed as distinct variables within the same datasets, ensuring that the two are not conflated and are interpreted separately.

5. Peripheral LMP Population:

The terminology regarding the "m+LMP population" is somewhat confusing. A clearer term might be "*Msx1*+ cells" as several clusters express *Msx1*, indicating a heterogeneous population. Also, it would be helpful to indicate which clusters are considered peripheral mesenchyme.

Agreed and we now refer throughout text and Figures to *Msx1*⁺ progenitors. There are four *Msx1*⁺ progenitor clusters that we identify in wildtype limb buds by gene enrichment analysis which results in definition of specific signatures for each cluster (**new Fig. 3a-d, Supplementary Tables 2, 3**). This gene enrichment analysis identifies the molecular identities of the four *Msx1* expressing clusters P1-P4. The dot plot and heatmap analysis show that cells expressing these signature genes are also detected in other clusters (**Fig. 3b, d**). Subsequent analysis combining DEGs with RNA-FISH (**Fig. 4,5**) identifies pLMPs as a peripheral (i.e. sub-ectodermal) mesenchymal population derived from *Msx1*⁺ progenitors. While wildtype pLMPs corresponds to most but not all cells in cluster P3, their signature is also detected the anterior-proximal cluster P1 and the autopodial-digit cluster P4 (see e.g. **Fig. 3b, 4e**).

6. pLMP Marker Genes:

The authors should explain why *Lhx2*, *Asb4*, *Msx2*, and *Rspo4* were chosen as markers for pLMP. For instance, *Msx2* and *Rspo4* are strong markers for clusters 3 and 10, while *Asb4* is expressed in several clusters, raising questions about their specific relevance.

The pLMP signature/marker genes were identified by comparative DEG analysis of wild-type versus both types of mutant forelimb buds. These DEGs are consistently up-regulated in the autopodial-digit (P4) and posterior (P2) clusters (**Fig. 4d**), while little to no changes are detected in the anterior clusters P2 and P1 (**Supplementary Fig. 5a, b**). In *Grem1* tetradactyl limb buds the number of cell expressing the pLMP score is significantly increased in P4 and P2 (**Fig. 4e**) in agreement with the posterior expansion observed by RNA-FISH analysis (**Fig. 5**).

7. Figure 4 and Supplementary Figure 4:

The added value of showing the differences in cell numbers of the pLMP population (Fig.4d) after demonstrating differences in cluster cell numbers (Fig.4a) is unclear.

Based on the additional analysis included in the revised **Fig. 4**, it is apparent that neither all pLMPs nor dLMPs are confined to a single cluster, nor are specific clusters composed exclusively of pLMPs or dLMPs. The bar plots in **Figs. 4e** and **Fig. 4g** illustrate the distribution and percentage of cells with positive pLMP and dLMP scores within the distal *Msx1*+progenitor clusters (anterior-P3, autopodial-P4 and posterior-P2). Statistically significant differences in pLMP and dLMP cell numbers between wild-type and *E1C8* mutants, as well as wild-type and *E1C5* mutants, are indicated in the bar plots by asterisks (p-value ≤ 0.001). This distribution is in agreement with the RNA-FISH analysis (**Figs. 5, 6 and Supplementary Figs 6, 7**).

pLMP population size is increased in distal and distal-posterior clusters"- the terminology (population size in clusters) is confusing. The data presented shows that in the mutants, the pLMP scores are increased in cells located in the distal and distal-posterior regions within the limb.

Agreed. We have clarified the terminology in the revised manuscript text.

" DEGs downregulated in cluster 8 is large restricted to the distal clusters" - Fig. 4a is not relevant to this point. It should probably be Fig. 4b.

As part of the complete revision of **Fig. 4** and the text describing these results, this has been fixed.

Please, provide precise panel references in Supp. Figure 4.

This is now **Supplementary Fig. 5** and all panel references are included.

8. dLMP Signature Genes:

The selection of *Tfap2b*, *Jag1*, and *Hoxa13* as signature genes for the distal-posterior LMP population raises questions about how these genes were chosen. These genes were previously used as markers for autopodial progenitors (Scotti et al., 2015, Zhang et al., 2023, Markman et al. 2023.). For the sake of clarity and uniformity, the authors might consider using this term.

The dLMP signature/marker score genes (*Tfap2b*, *Jag1*, *Hoxa13* and *Hoxd13*) were identified by comparative DEG analysis of wild-type versus both types of mutant forelimb buds. While these DEGs are indeed part of the gene enrichments signature of the autopodial-digit (P4) cluster, dLMPs are also present in the posterior cluster (P2), but represent only a subset of all wild-type progenitor cells in both clusters (**Fig. 4g**). In both types of mutant forelimb buds, dLMP cell numbers are consistently down-regulated in the autopodial-digit (P4) and posterior (P2) clusters, whereas little or no changes are detected in the anterior clusters P2 and P1 (**Fig. 4g, Supplementary Fig. 5a, b**).

9. Both LMP populations originate from m+LMPs(38):

Please clarify. To my understanding, dLMPs are equivalent to "Autopodial progenitors" in the cited paper, while pLMPs are likely equivalent to "Naïve progenitors".

As described in the response to points 6-8 above, the distinct signatures of dLMPs and pLMPs were identified in an unbiased manner by DEG analysis of wildtype versus tetradactyl mutant forelimb buds prior to apparent autopod formation. These gene expression profiles indeed exhibit similarities with previously published autopodial signatures of wildtype limb bud transcriptome atlases, which we discuss in the text (**ref. 11, 39, 49, 78 in the manuscript**).

Previous lineage analysis for two dLMP markers shows that *Hoxa13* gives rise to autopodial lineages (**ref. 12**), while *Tfap2b* might be a true digit lineage (**ref. 53**). In addition, *Jag1* expression marks the distal autopod mesenchyme (**ref. 39**). Our comparative analysis of wild-type and *Grem1* tetradactyl forelimb buds indicates that that dLMPs are a subset of the autopodial progenitors described in **ref. 11** (see point 8).

pLMPs: likewise, this population has a similar but not identical signature to the naïve progenitors described by Markman *et al.* 2023. For example, *Lmo2* is part of the naïve progenitor signature but not of pLMPs. Therefore, we have included the following statement in the first paragraph of the **Discussion** section with respect to the potential origin of the two LMP populations: "These two distinct LMP populations exhibit gene expression profiles that in large parts overlap the molecular signature of *Msx1*+ progenitors characterized by an autopodial genetic program identified previously in wildtype limb buds¹¹."

10. *Msx1* Expression in Mutants:

The authors claim that the *Msx1* expression domain in the mutant limb is broader than in the control (Fig. 5a). However, the figure does not clearly represent these differences.

Additionally, *Msx1* expression levels between WT and mutants seem inconsistent between the left and the right panels in Fig.5a.

Agreed- the *Msx1* expression domain is broader in the distal and posterior regions in both mutant forelimb buds in comparison to wild-type limb buds. We have replaced the *Sox9/Msx1* co-localisation in the right panels of **Fig. 5a** and include the individual black and white channels in **Supplementary Fig. 6a**. In addition, we include a figure below to illustrate the variations in the spatial *Msx1* expression detected. In the text we also state that there is some variation.

11. pLMP and dLMP Spatial Distribution:

To accurately demonstrate the location of pLMPs using the selected markers, the authors should provide in situ evidence showing the overlap of *Lhx2* and *Asb4*. Similarly, demonstrating the location of dLMPs (*Jag1* and *Tfap2b*) also necessitates in situ evidence of overlap between the chosen markers.

Agreed and done: the results of the extended RNA-FISH analysis with single black and white channels (see also below) are included in the revised **Fig. 5** and new **Fig. 6** and **Supplementary Figs 6, 7**.

12. Figure 5 (Salt-and-Pepper Pattern):

The current images do not adequately support the "salt and pepper" pattern claim. Increasing resolution and magnification is necessary, along with quantitative validation of dLMP cell numbers and gene expression levels. Providing single-channel images for Figures 5e-f would also be beneficial.

Agreed. Single channel images are included as part of the main **Fig. 5, 6** and **Supplementary Fig. 6, 7**. higher magnification analysis of optical sections shows that the *Jag1*-expressing dLMP domain is reduced in mutant forelimb buds while the interspersed *Lhx2*-expressing pLMPs are increased. This finding is clear and does indeed not require nor fit a "salt and pepper" pattern (**Fig. 5d**).

We include single cell scatterplots showing the cells co-expressing the marker genes analysed by RNA-FISH. These scatterplots show the statistically significant reductions in both mutants, which is totally with the spatial reductions observed by RNA-FISH (**Fig. 6 and Supplementary Fig. 7**). In addition, the posterior expansion of pLMPs expression observed by RNA-FISH is consistent with the bar plot analysis that shows the increase of pLMP cells in the autopodial and posterior cluster in both mutants (**Fig. 4e**). Therefore the spatial alterations observed by RNA-FISH analysis are consistent with the statistically significant cellular changes.

RNA-FISH analysis of the spatial *Msx1* distribution in wildtype and *Grem1* tetradactyl forelimb buds

Fig. 5a (n=6 biological replicates)

Statement in the **Results** section: "While *Msx1* expression is anteriorly biased in wildtype limb buds, its spatial domain broadens in the distal and posterior mesenchyme in *E1C5*^{Δ/Δ} and *E1C8*^{Δ/Δ} limb buds"